# *Nr4a1* suppresses cocaine-induced behavior via epigenetic regulation of homeostatic target genes

Marco D. Carpenter [1,2,3], Qiwen Hu[1,2,3], Allison M. Bond[3,4], Sonia I. Lombroso[1,2,3], Kyle S. Czarnecki[1,2,3], Carissa J. Lim[1], Hongjun Song[3,4], Mathieu E. Wimmer[5], R. Christopher Pierce[6] & Elizabeth A. Heller[1,2,3]*

Endogenous homeostatic mechanisms can restore normal neuronal function following cocaine-induced neuroadaptations. Such mechanisms may be exploited to develop novel therapies for cocaine addiction, but a molecular target has not yet been identified. Here we profiled mouse gene expression during early and late cocaine abstinence to identify putative regulators of neural homeostasis. Cocaine activated the transcription factor, *Nr4a1*, and its target gene, *Cartpt*, a key molecule involved in dopamine metabolism. Sustained activation of *Cartpt* at late abstinence was coupled with depletion of the repressive histone modification, H3K27me3, and enrichment of activating marks, H3K27ac and H3K4me3. Using both CRISPR-mediated and small molecule *Nr4a1* activation, we demonstrated the direct causal role of *Nr4a1* in sustained activation of *Cartpt* and in attenuation of cocaine-evoked behavior. Our findings provide evidence that targeting abstinence-induced homeostatic gene expression is a potential therapeutic target in cocaine addiction.

[1] Department of Systems Pharmacology and Translational Therapeutics, University of Pennsylvania, Philadelphia, PA 19104, USA. [2] Institute for Translational Medicine and Therapeutics, University of Pennsylvania, Philadelphia, PA 19104, USA. [3] Penn Epigenetics Institute, Perelman School of Medicine, University of Pennsylvania, Philadelphia, PA 19104, USA. [4] Department of Neuroscience, University of Pennsylvania, Philadelphia, PA 19104, USA. [5] Department of Psychology and Program in Neuroscience, Temple University, Pennsylvania, Philadelphia, PA 19122, USA. [6] Center for Nurobiology and Behavior, Department of Psychiatry, Perelman School of Medicine at the University of Pennsylvania, Philadelphia, PA 19104, USA. *email: eheller@pennmedicine.upenn.edu

A core feature of addiction is the propensity for relapse due to the aggregation of neuroadaptations during abstinence[1]. In human drug users, relapse is triggered by drug-associated cues[2], stress[3], and acute drug re-exposure[4]. To combat relapse, endogenous homeostatic mechanisms may restore and even reverse normal function to reward-related brain areas[5,6]. Both preclinical[7] and human[8] studies indicate that drug-induced synaptic plasticity and the associated drug memories are reversible. Mechanisms that mediate adaptive changes to reduce drug-seeking behavior include gene expression during cocaine abstinence[9]. Therefore, the identification of a molecular mechanism of sustained homeostatic transcription across cocaine abstinence has important therapeutic implications. However, only a limited number of studies have profiled global gene expression both immediately following drug taking and after abstinence[10,11].

Many brain regions are involved in the formation of drug-associated neuroadaptations, including the ventral tegmental area (VTA), prefrontal cortex (PFC), and nucleus accumbens (NAc)[11]. To identify a master regulator of homeostatic gene expression, we profiled global transcriptomic changes in the NAc, VTA, and PFC at early and late abstinence following cocaine self-administration in mice. Using this approach, we identified a key role for the transcription factor, nuclear receptor subfamily 4 group A member 1 (Nr4a1), in regulating homeostatic target gene expression and cocaine-evoked behavior. Nr4a1 plays an integral role in neuronal homeostasis and neuroprotection in response to hyperexcitation via the regulation of downstream effectors contributing to synapse distribution and function[11–13]. In response to stimuli, Nr4a1 is upregulated and it is shuttled to the nucleus where it binds nerve-growth-factor inducible gene B (NGFI-B)-responsive elements (NRBE) at target gene promoters[14]. Importantly, Nr4a1 regulates gene expression via recruitment of chromatin modifying enzymes, many of which are stable across abstinence[14,15].

Histone modifications persist at specific genomic loci during abstinence and play an important role in stable transcriptional regulation associated with addictive behaviors[9]. Given that chromatin modifications confer long-lasting changes in gene expression necessary for stable cellular phenotypes, histone modifications acquired during abstinence may cause individual genes to "remember" prior drug exposure. Indeed, Nr4a1 is transiently expressed during learning and supports memory formation via histone acetylation and activation of downstream target genes in the hippocampus[16]. Loss of Nr4a1 activation causes deficits in long-term potentiation, abnormal increases in spine density and impaired long-term memory[12,16,17]. Beyond this, altered levels of Nr4a1 and Nr4a2 expression are associated with Parkinson's disease[18], schizophrenia[19], and cocaine addiction in humans[20,21], due to its function in CREB-mediated neuroprotection and dopamine-related neuroadaptation[22,23]. Nr4a1 is highly expressed in striatal regions of dopaminergic output, such as the NAc, where it determines striatal dopamine levels[22] via activation of target genes including cocaine and amphetamine-regulated transcript peptide (Cartpt)[24]. Nr4a1-deficient mice show increased sensitivity to psychostimulants measured by amphetamine-induced locomotor activity and dopamine metabolism[25,26]. In fact, our findings are consistent with prior studies that show cocaine-induced expression of Nr4a1[10,11,27].

We identified Nr4a1 and its target genes as potentially important mediators of homeostatic gene expression across abstinence using an unbiased transcriptomic approach. We prioritized Nr4a1 given its key roles long-term memory and neuroprotection. We discovered a mechanism whereby Nr4a1 stably regulated key histone modifications and activated target genes involved in neuronal homeostasis, including Cartpt.

Importantly, activation of Nr4a1 reduced cocaine reinforced behavior. Herein, we established Nr4a1 as a key regulator of persistent gene transcription during cocaine abstinence and as a promising therapeutic target for cocaine addiction.

## Results

**Cocaine regulated Nr4a1 via histone modifications (hPTMs).** Several studies suggest that drug exposure increases homeostatic gene expression during abstinence to mitigate cocaine induced neuroadaptations[5]. To identify a master regulator of homeostatic gene expression, we profiled global transcriptomic changes in the adult mouse brain, including the NAc, VTA, and PFC, at early (1-day) and late (28-days) abstinence following cocaine self-administration (Fig. 1a). In all cases, we compared cocaine to saline treated tissue at each time point. All cocaine treated mice acquired self-administration behavior (SA), measured by a greater number of infusions, active (cocaine-paired) spins and discrimination between the active and inactive (saline-paired) wheels across 21 daily sessions (Fig. 1b; Supplementary Fig. 1A–F). In the NAc, we found a greater number of regulated transcripts at 28-days (341 differentially expressed genes (DEGs)) than at 1-day (44 DEGs) of abstinence (Fig. 1c, d)[10]. Alternatively, in the VTA and PFC there were a greater number of cocaine-regulated genes at 1-day (DEGs: VTA 3040, PFC 82) than at 28-days (DEGs: VTA 1571, PFC 45) of abstinence (Supplementary Fig. 2A–F). Nr4a1 was identified in the highest-ranked biological process group, cellular response to the corticotropin-releasing hormone, using Gene ontology (GO) analysis on the NAc DEGs (Fig. 1e, top). Interestingly, at 28-days of abstinence there was enrichment for the biological process of memory, which includes genes involved in the acquisition, modification, and retrieval of informational stimuli (Fig. 1e, bottom). RNA-seq measured activation of Nr4a1 at 1-day but not 28-days of abstinence (Fig. 1f). Nr4a1 activation and several other DEGs were validated via qPCR in a separate cohort of animals that underwent the SA paradigm (Fig. 1g; Supplementary Figs. 3A–D, 4A–D; Supplementary Table 1). Nr4a1 is activated in brain as an adaptive response to hyperexcitation, and functions as a transcription factor to regulate target genes necessary for synaptic reorganization[12,13,28]. We found no significant differences in the activation of Nr4a1 in the VTA or PFC at either 1-day or 28-days of abstinence, suggesting a NAc-specific mechanism of Nr4a1-dependent transcription across abstinence.

To interrogate the mechanism of Nr4a1 gene activation, we measured the enrichment of Nr4a1 protein and the hPTMs, histone H3 lysine 27 trimethylation (H3K27me3) and acetylation (H3K27ac), and histone H3 lysine 4 trimethylation (H3K4me3) at the Nr4a1 promoter following repeated, investigator-administered cocaine (Fig. 1h). The technical ease and reduced variability of this treatment paradigm relative to cocaine SA allowed us to profile transcription factor binding and multiple epigenetic modifications at several loci. We first validated Nr4a1-binding to the promoters of cocaine-activated genes in the NAc by quantitative chromatin immunoprecipitation (qChIP) using an Nr4a1 specific antibody (Supplementary Fig. 5A; Supplementary Table 2). As in cocaine SA, repeated investigator administered cocaine activated Nr4a1 at 1-day (Fig. 1i) and increased the enrichment of Nr4a1 at the Nr4a1 promoter (Fig. 1j) in the NAc at 1-day and 28-days of abstinence. We found no differences in the repressive modification, H3K27me3 (Fig. 1k), and enrichment of the activating hPTMs, H3K27ac (Fig. 1l) and H3K4me3 (Fig. 1m) between cocaine and saline after 28-days of abstinence. Taken together, these findings indicated that cocaine induced transient activation of Nr4a1 and persistent changes in chromatin state at the Nr4a1 promoter.

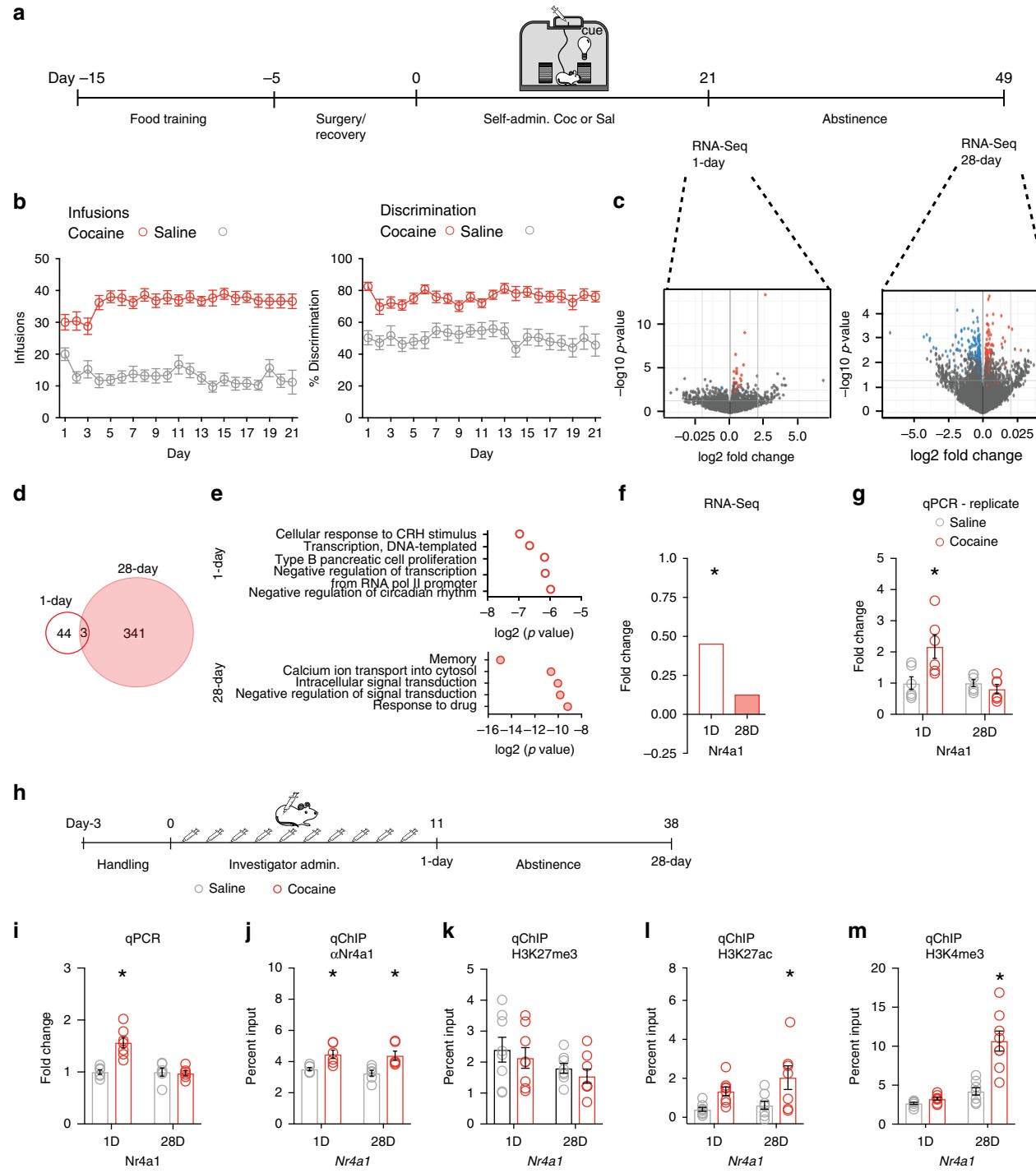

**Cocaine regulated Nr4a1 target genes via hPTMs.** To determine the mechanism of action of *Nr4a1* across abstinence, we measured expression of known Nr4a1 target genes that were regulated by cocaine abstinence and are relevant in the neuroadaptive processes underlying drug-associated behavior. Specifically, we found that Nr4a1 bound to the promoter region of (1) casitas B-lineage lymphoma (*Cbl*), an E-3 ubiquitin-ligase that regulates Nr4a1 expression via targeting to the proteasome[29] (2) period circadian regulator 2 (*Per2*), which is activated by cocaine in the striatum and negatively regulates cocaine sensitization and conditioned place preference (CPP)[30], and (3) *Cartpt*, which is increased in the NAc of both humans and mice during abstinence from cocaine[10,31,32] (Supplementary Fig. 5A). With respect to

cocaine-activation of these Nr4a1 target genes, RNA-seq measured transient activation of *Cbl* at 1-day of abstinence and no significant difference at 28-days of abstinence (Fig. 2a), while *Per2* was activated at 1-day and repressed at 28-days of abstinence (Fig. 2a). *Cartpt* was activated only at 28-days of abstinence (Fig. 2a). These expression patterns were validated in a distinct cohort of mice via qPCR (Fig. 2b).

We hypothesized that Nr4a1 enrichment facilitates the recruitment of histone modifying enzymes at target genes to regulate long-lasting changes in transcription[15,33]. To test this, we measured the enrichment of Nr4a1 at *Cartpt*, a gene with delayed activation after 28-days of abstinence. Similar to cocaine SA, repeated investigator administered cocaine activated *Cartpt*

**Fig. 1 Cocaine regulated activation of *Nr4a1* and promoter function. a** Timeline of cocaine self-administration (SA) (0.7 mg/kg/inf, 21 days) and abstinence (1-day and 28-days). **b** Cocaine SA mice responded significantly higher than saline self-administering mice (left, $n = 24$ mice/group, two-way repeated measures ANOVA) and discriminated between active and inactive wheels, while saline mice responded equally (right, $n = 24$ mice/group, two-way repeated measures ANOVA) during 21-days of self-administration. *$P < 0.05$. **c** Volcano plot showing differentially expressed genes (DEGs), significantly downregulated (Blue) and upregulated (Red); $q$-value $< 0.01$ **d** Comparison of DEGs at 1-day and 28-days of abstinence. **e** Gene ontology (GO) analysis of DEGs at 1-day (Top) and 28-days (Bottom) of abstinence. $P < 0.05$. **f** *Nr4a1* activation at 1-day of abstinence following cocaine SA measured by RNA-seq log2 fold change, $q$-value $< 0.01$. **g** *Nr4a1* expression measured by qPCR in a biological replicate ($n = 6$ mice/group, two-way ANOVA, followed by Tukey's multiple comparisons test, $P = 0.0097$). *$P < 0.05$. **h** Timeline of repeated investigator-administered (IA) cocaine (20 mg/kg; i.p.) and abstinence (1-day and 28-days). **i** Cocaine activated Nr4a1 at 1-day ($P = 0.0001$) but not 28-days of abstinence ($P = 0.9967$), relative to saline controls ($n = 6$-7 mice/group, two-way ANOVA, followed by Tukey's multiple comparisons test). *$P < 0.05$. **j** Cocaine enriched Nr4a1 at the *Nr4a1* promoter at 1-day ($P = 0.0343$) and 28-days ($P = 0.0101$) of abstinence, normalized to input, relative to saline controls ($n = 6$ mice/group, two-way ANOVA, followed by Tukey's multiple comparisons test). *$P < 0.05$. **k** Cocaine had no effect on H3K27me3 enrichment at either 1-day or 28-days of abstinence, normalized to input, relative to saline controls ($n = 8$ mice/group, two-way ANOVA, followed by Tukey's multiple comparisons test $P = 0.0527$) *$P < 0.05$. **l** Cocaine enriched H3K27ac at the *Nr4a1* promoter at 28-days ($P = 0.0096$), but not 1-day ($P = 0.0916$) of abstinence, normalized to input, relative to saline controls ($n = 7$-8 mice/group, two-way ANOVA, followed by Bonferroni's multiple comparisons test). *$P < 0.05$. **m** Cocaine enriched H3K4me3 at the *Nr4a1* promoter at 28-days ($P < 0.0001$) but not 1-day ($P = 0.9363$) of abstinence, normalized to input, relative to saline controls ($n = 8$ mice/group, two-way ANOVA, followed by Tukey's multiple comparisons test). *$P < 0.05$. All error bars represent mean $+/-$ s.e.m. Source data and statistics provided as a Source Data file.

specifically at 28-days, but not 1-day, of abstinence (Fig. 2c). Nr4a1 was enriched at *Cartpt* at 28-days of abstinence (Fig. 2d). To investigate the regulatory mechanism for delayed activation of *Cartpt*, we measured hPTMs at the promoter region. At 1-day of abstinence, at which point *Cartpt* is repressed, we measured enrichment of the repressive hPTM, H3K27me3 (Fig. 2e), as well as enrichment of the activating hPTMs, H3K27ac (Fig. 2f) and H3K4me3 (Fig. 2g). *Cartpt* was activated only at late abstinence (28-days), along with H3K27me3 depletion (Fig. 2e) and H3K27ac enrichment (Fig. 2f); H3K4me3 was also enriched at *Cartpt* at this time point (Fig. 2g). This finding suggested abstinence-induced activation of *Cartpt* was associated with sustained Nr4a1 enrichment and depletion of the repressive hPTM, H3K27me3. Therefore, Nr4a1 may act as a master regulator of delayed cocaine-mediated gene activation via sustained binding at target genes.

**CRISPR regulation of *Nr4a1* and target genes**. We next applied CRISPR-activation to determine the causal relevance of *Nr4a1* to homeostatic target gene expression during cocaine abstinence. This system consists of 2 components: (1) a catalytically dead Cas9 protein fused to a VP64 transcriptional activation domain (dCas9-VP64), driven by a neuron specific promoter and (2) a single-guide RNA (sgRNA) targeting the *Nr4a1* promoter (Supplementary Table 3). dCas9-VP64 moderately activated gene expression which is advantageous in obtaining physiologically relevant levels of *Nr4a1* activation[34]. For control experiments, dCas9-VP64 was paired with a control, non-targeting sgRNA (sgRNA-NT) with no homology to the mouse genome (Fig. 3a). We first used Neuro2a (N2a) cells to screen a library of sgRNAs that targeted the *Nr4a1* promoter at various locations upstream ($-366$, $-231$, $-36$ bps) or downstream ($+82$, $+200$ bps) of the transcription start site (TSS) (Fig. 3b). *Nr4a1*-sgRNA-366 and *Nr4a1*-sgRNA-231 both activated Nr4a1 mRNA expression, relative to control sgRNA-NT (Fig. 3b). We also measured activation of *Cartpt*, as it is associated with Nr4a1 binding during cocaine abstinence (Fig. 3c). To repress *Nr4a1* we utilized dCas9 fused to KRAB, a transcriptional repressor previously validated in vivo[35]. In N2a cells, co-transfection of *Nr4a1* sgRNAs with dCas9-KRAB did not change Nr4a1 expression (Fig. 3b).

To examine potential off-target effects, we performed engineered DNA-binding molecule ChIP (enChIP), which allows qChIP with an antibody that recognizes dCas9 fused to a synthetic affinity tag (AMtag)[36]. When co-transfected with sgRNA-231, dCas9-AMtag was enriched above input only at

the sgRNA-*Nr4a1* target region and not at a distal site -1000 bps from the *Nr4a1* TSS (Fig. 3d). There was no enrichment above input when dCas9-AMtag was transfected with sgRNA-NT (Fig. 3d).

We next examined the feasibility of CRISPR-mediated *Nr4a1* activation and repression in vivo, by co-transfection of sgRNA-*Nr4a1* (sgRNA-231) and dCas9-VP64 or dCas9-KRAB in the NAc. In our stereotaxic transfections, expression is primarily in the core sub region of the NAc, with small amounts of expression in shell. In all cases, sgRNA-*Nr4a1* and sgRNA-NT were injected contralaterally, to accomplish rigorous, within-animal comparisons. We first confirmed neural specificity of dCas9-VP64, conferred by the synapsin (hSyn) promoter, via double immunohistochemical analysis using anti-Cas9 and anti-NeuN antibodies. We found that dCas9-VP64 colocalizes exclusively with NeuN positive neurons in the NAc, 24 h post-transfection (Fig. 3e; Supplementary Fig. 6A). By qPCR, co-transfection of sgRNA-*Nr4a1* and dCas9-VP64 activated *Nr4a1*, relative to sgRNA-NT (Fig. 3f). Co-transfection of sgRNA-*Nr4a1* with dCas9-KRAB repressed *Nr4a1* (Fig. 3f). We performed several control experiments to evaluate specificity of this bidirectional effect on mRNA expression. We found that dCas9-VP64 nor dCas9-KRAB in combination with sgRNA-NT, or dCAs9-VP64 nor sgRNA alone or *Cdk5*-sgRNA (an unrelated gene), were sufficient to regulate activation of *Nr4a1* (Fig. 3f). We next determined the expression of Nr4a1 target genes and found that sgRNA-*Nr4a1* co-transfected with dCas9-VP64 or dCas9-KRAB caused activation or repression, respectively, of *Cartpt*, relative to sgRNA-NT (Fig. 3g, h). We determined the time course of CRISPR-mediated *Nr4a1* activation and found activation of *Nr4a1* and *Cartpt* at 4-days and 7-days, but not 14-days post transfection (Fig. 3i). Finally, as in cocaine SA, CRISPR-mediated activation of *Nr4a1* depleted H3K27me3 (Fig. 3j) and enriched H3K27ac (Fig. 3k). There were no significant changes in H3K4me3 (Fig. 3l). We conclude *Nr4a1* activation coordinates the depletion of H3K27me3 and enrichment of H3K27ac to facilitate *Cartpt* activation.

**Nr4a1 bidirectionally modulated cocaine behavior**. Having established the role of *Nr4a1* in mediating gene transcription and hPTMs at cocaine-induced target genes, we next determined the effect of CRISPR-mediated *Nr4a1* regulation on cocaine-induced behavior and gene expression. These effects can be evaluated in mouse place conditioning paradigms or self-administration. Because the time course of conditioned place preference (CPP)

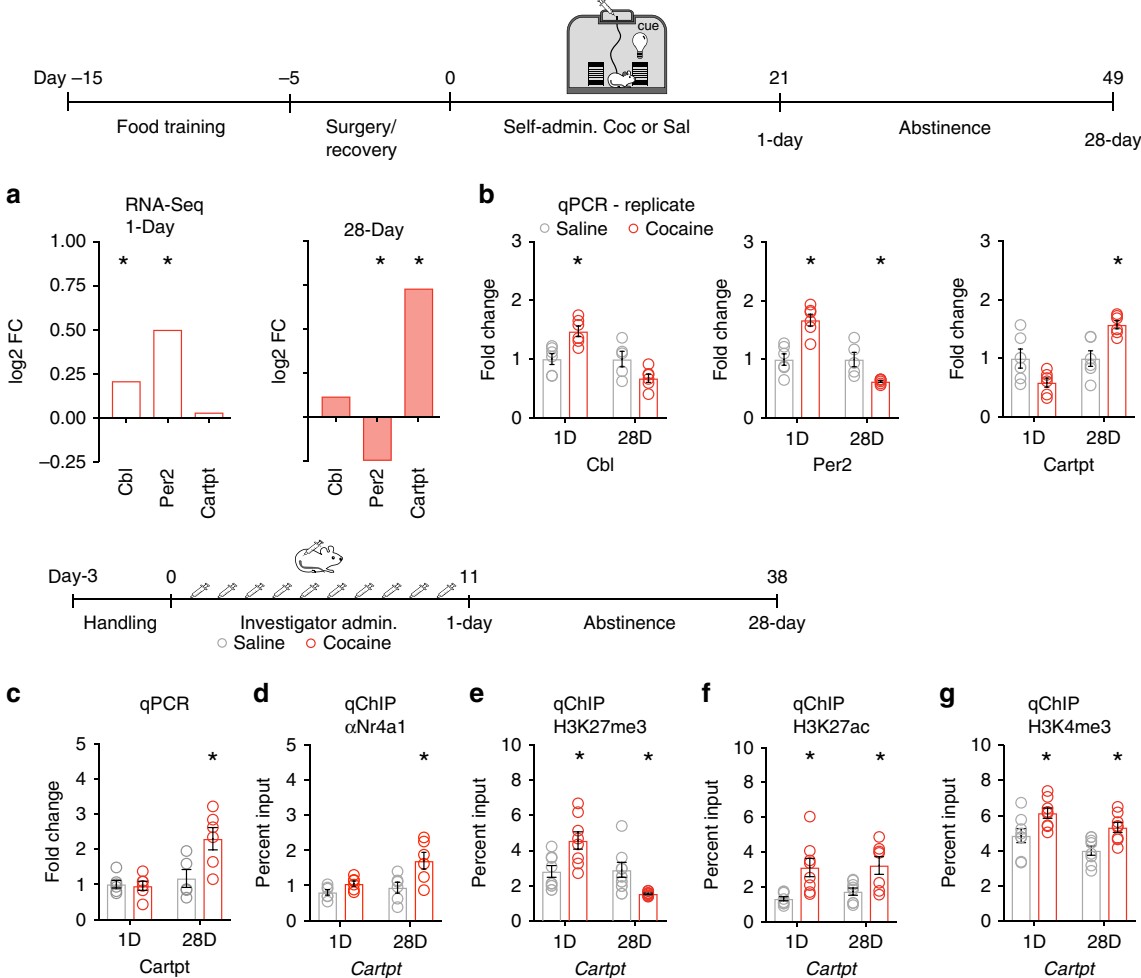

**Fig. 2 Cocaine regulated *Nr4a1* target genes and promoter function. a** RNA-seq measures activation of *Cbl* and *Per2* following cocaine SA at 1-day of abstinence and *Cartpt* activation and *Per2* repression at 28-days, log2 fold change normalized to saline controls, q-value < 0.001. **b** qPCR following cocaine SA, in a biological replicate, identifies activation of *Cbl* ($n = 5-6$ mice/group, two-way ANOVA, followed by Tukey's multiple comparisons test, 1-day $P = 0.0099$) and *Per2* ($n = 5-6$ mice/group, two-way ANOVA, followed by Tukey's multiple comparisons test, 1-day $P = 0.0002$) at 1-day of abstinence, no change in *Cbl* ($P = 0.1197$) and repression of *Per2* ($P = 0.046$) at 28-days of abstinence, and *Cartpt* activation at 28-days ($n = 6$ mice/group, two-way ANOVA, followed by Tukey's multiple comparisons test, 28-days $P = 0.0112$) of abstinence but not at 1-day ($P = 0.0934$) of abstinence. *$P < 0.05$. **c** qPCR following repeated cocaine identifies *Cartpt* activation at 28-days ($n = 6$ mice/group, two-way ANOVA, followed by Tukey's multiple comparisons test, 28-days $P = 0.0095$ of abstinence but not at 1-day $P = 0.9987$ of abstinence relative to saline controls. *$P < 0.05$. **d** Cocaine enriched Nr4a1 at the *Cartpt* promoter at 28-days ($n = 6$ mice/group, two-way ANOVA, followed by Tukey's multiple comparisons test, 28-days $P = 0.0089$) but not at 1-day of abstinence ($P = 0.6515$), normalized to input, relative to saline controls. *$P < 0.05$. **e** Cocaine enriched H3K27me3 at the *Cartpt* promoter at 1-day ($n = 8$ mice/group, two-way ANOVA, followed by Tukey's multiple comparisons test, 1-day $P = 0.0112$) of abstinence and depleted H3K27me3 from the *Cartpt* promoter at 28-days ($P = 0.0381$) of abstinence, normalized to input, relative to saline controls. *$P < 0.05$. **f** Cocaine enriched H3K27ac at the *Cartpt* promoter 1-day and 28-days of abstinence ($n = 7-8$ mice/group, two-way ANOVA, followed by Tukey's multiple comparisons test, 1-day $P = 0.0097$, 28-days $P = 0.0461$), normalized to input, relative to saline controls. *$P < 0.05$. **g** Cocaine enriched H3K4me3 at the *Cartpt* promoter at 1-day and 28-days of abstinence ($n = 7-8$ mice/group, two-way ANOVA, followed by Tukey's multiple comparisons test, 1-day $P = 0.0367$, 28-days $P = 0.0297$), normalized to input, relative to saline controls. *$P < 0.05$. All error bars represent mean $+/-$ s.e.m. Source data and statistics provided as a Source Data file.

matched more precisely that of CRISPR-mediated *Nr4a1* activation, we first examined the effects in this paradigm. Most drugs that are self-administered also produce CPP suggesting there may be similar molecular substrates underlying drug-memories and drug-seeking[37]. As shown schematically, bilateral NAc of each subject was co-transfected with dCas9-VP64 and sgRNA-*Nr4a1* or -NT (control) before CPP was performed; the effect of *Nr4a1* repression by dCas9-KRAB was examined in a separate cohort (Fig. 4a).

Prior to cocaine conditioning, both control mice and those with activated *Nr4a1* showed no differences in their preference for the saline-paired chamber (Fig. 4b). Following conditioning, both groups of mice spent more time in the cocaine-paired chamber, compared to the pre-test (Fig. 4b), indicating that activation of *Nr4a1* did not perturb the ability of mice to form a preference for cocaine. We found that cocaine preference was attenuated in mice following CRISPR activation of *Nr4a1*, compared to control mice. A separate cohort receiving a higher dose of cocaine showed a similar effect, that is, reduced cocaine CPP following *Nr4a1* activation (Fig. 4b). At both doses, this attenuation of CPP was accompanied by activation of *Cartpt* (Fig. 4c). Alternatively, CRISPR repression of *Nr4a1* increased preference for the cocaine-paired chamber in mice administered a low dose of cocaine at which control mice did not form cocaine

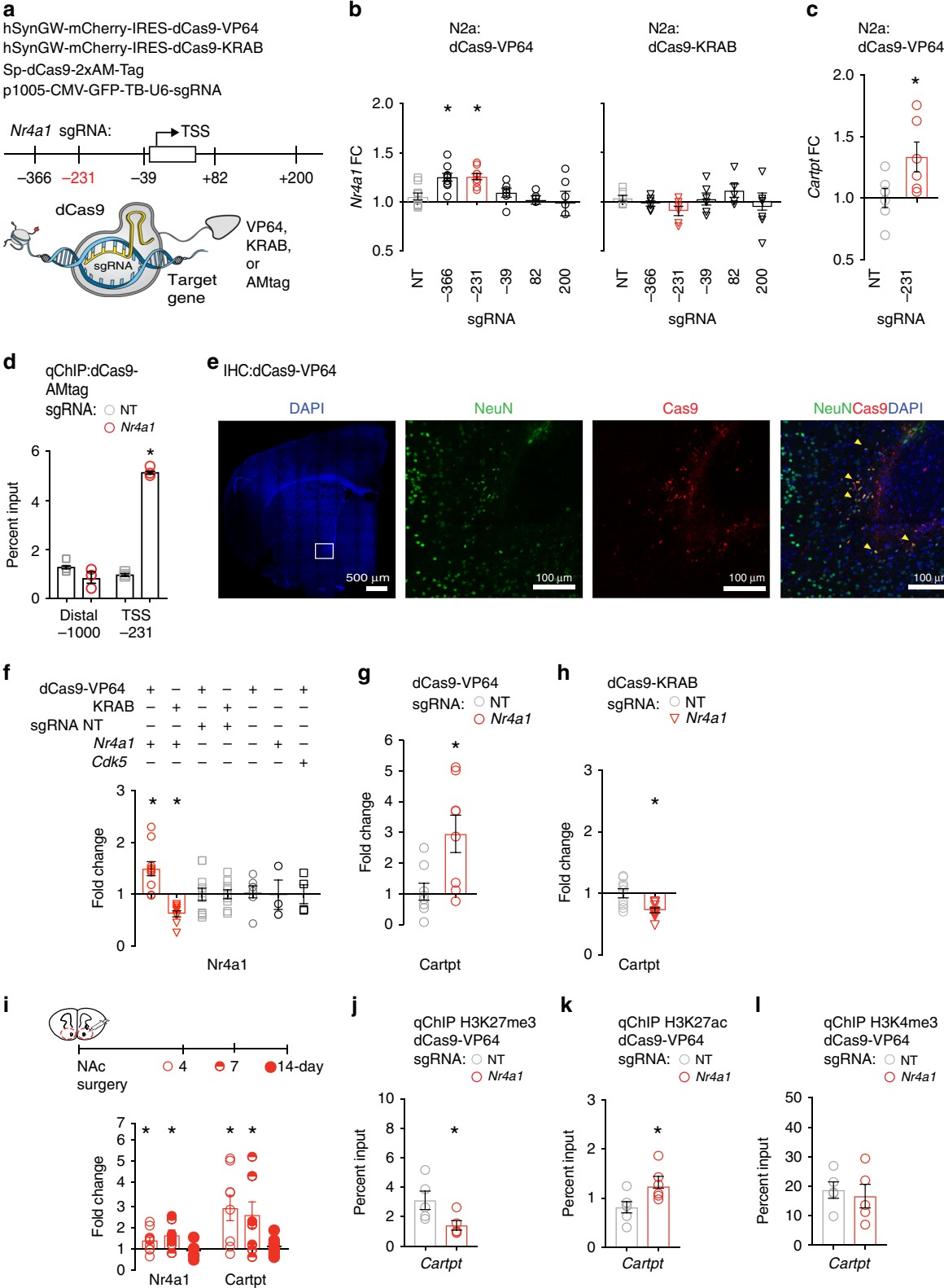

CPP (Fig. 4d). At a dose to which control mice develop a preference, *Nr4a1* repression also increased cocaine CPP (Fig. 4d). Increased cocaine preference was associated with repression of *Cartpt* at both doses (Fig. 4e). To determine if deficits in baseline locomotor activity could account for these effects, total distance traveled was measured and no significant differences between groups were found (Supplementary Fig. 7A–D).

We next investigated the role of *Nr4a1* in acquisition of cocaine SA and cocaine-seeking behavior during abstinence (Fig. 4j–l). In cocaine-seeking, mice freely perform the operant tasks in the presence of cues, but no reinforcer is given for active responses. CRISPR activation of *Nr4a1* prior to cocaine SA had no effect on acquisition (Fig. 4g; Supplementary Fig. 8A–D) or cocaine-seeking at 1-day of abstinence (Fig. 4h; Supplementary Fig. 8E–G).

**Fig. 3 CRISPR regulation of *Nr4a1* and target genes in vitro and in vivo. a** Neuronal expression of dCas9 via synapsin promoter (hSyn). Schematic of five sgRNAs relative to the transcription start site (TSS). **b** sgRNA-231 and dCas9-VP64 activated *Nr4a1* in N2a cells ($n = 6$–11 mice/group, one-way ANOVA, followed by Bonferroni's multiple comparisons test, NT vs. −231 $P = 0.0123$, NT vs. −366 $P = 0.0148$), normalized to GAPDH, and sgRNA-NT. *$P < 0.05$. **c** sgRNA-231 and dCas9-VP64 activated *Cartpt* in N2a cells (unpaired two-tailed *t*-test, $P = 0.0411$) normalized to GAPDH, and sgRNA-NT. *$P < 0.05$. **d** dCas9 and sgRNA-231 bind specifically at the Nr4a1 promoter ($n = 3$ mice/group, two-way ANOVA, followed by Tukey's multiple comparisons test $P < 0.0001$), but not at distal site ($P = 0.2338$) normalized to input, relative to sgRNA-NT. *$P < 0.05$. **e** dCas9-VP64 colocalizes with NeuN+ cells in the NAc. Yellow arrow heads indicate NeuN+/dCas9-VP64+ cells. DAPI = Blue, dCas9 = Red, Ne*u*n = Green, Anterior Commissure (AC). **f** sgRNA-*Nr4a1*/dCas9-VP64 activated and sgRNA-*Nr4a1*/dCas9-KRAB repressed *Nr4a1* in vivo ($n = 4$–11 mice/group, one-way ANOVA, followed by Bonferroni's multiple comparisons test, sgRNA-NT/dCas9-VP64 vs. sgRNA-*Nr4a1*/dCas9-VP64 $P = 0.0039$, sgRNA-NT/dCas9-KRAB vs. sgRNA-*Nr4a1*/dCas9-KRAB $P = 0.0483$), normalized to GAPDH, and sgRNA-NT. *$P < 0.05$. **g** sgRNA-*Nr4a1*/dCas9-VP64 activated *Cartpt* in vivo ($n = 8$, unpaired two-tailed *t* test, $P = 0.0138$), normalized to GAPDH, and sgRNA-NT. *$P < 0.05$. **h** sgRNA-*Nr4a1*/dCas9-KRAB repressed *Cartpt* in vivo ($n = 9$–10, unpaired two-tailed *t*-test, $P = 0.0033$), normalized to GAPDH, and sgRNA-NT. *$P < 0.05$. **i** dCas9-VP64 and sgRNA-*Nr4a1* activated *Nr4a1* and *Cartpt* at 4-days and 7-days but not at 14-days (*Nr4a1*: $n = 7$–10 mice/group, two-way ANOVA, followed by Bonferroni's multiple comparisons test, 1-day $P = 0.0225$, 4-days $P = 0.0318$, 14-days $P > 0.9999$; *Cartpt*: $n = 7$–10 mice/group, two-way ANOVA, followed by Bonferroni's multiple comparisons test, 1-day $P = 0.0094$, 4-days $P = 0.0359$, 14-days $P > 0.9999$), normalized to GAPDH, relative to sgRNA-NT. *$P < 0.05$. **j** dCas9-VP64 and sgRNA-*Nr4a1* depleted H3K27me3 at the *Cartpt* promoter ($n = 5$, unpaired two-tailed *t*-test, $P = 0.0421$), normalized to input, relative to saline controls. *$P < 0.05$. **k** dCas9-VP64 and sgRNA-*Nr4a1* enriched H3K27ac at the *Cartpt* promoter ($n = 6$, unpaired two-tailed *t*-test, $P = 0.0280$), normalized to input, relative to saline controls. *$P < 0.05$. **l** dCas9-VP64 and sgRNA-*Nr4a1* caused no significant change in H3K4me3 at the *Cartpt* promoter ($n = 5$, unpaired two-tailed *t*-test, $P = 0.6851$), normalized to input, relative to saline controls. *$P < 0.05$. All error bars represent mean $+/-$ s.e.m. Source data and statistics provided as a Source Data file.

Given the time course of homeostatic target gene activation following CRISPR activation of *Nr4a1*, we next activated *Nr4a1* during abstinence following cocaine SA, to coincide with a test of cocaine-seeking in late abstinence (Fig. 4i–k; Supplementary Fig. 9A–D). This manipulation attenuated cocaine-seeking at 28-days of abstinence (Fig. 4j; Supplementary Fig. 9E–G), at a time point when *Cartpt* is activated by cocaine SA (See Fig. 2a, b). In conclusion, *Nr4a1* activation is sufficient to mediate cocaine-dependent neuroadaptations associated with cocaine-induced behavior.

**Csn-B activated *Nr4a1* and attenuated cocaine behavior**. Given our findings that *Nr4a1* activation attenuated cocaine CPP and cocaine-seeking following abstinence, we investigated the therapeutic potential of its pharmacological activation. Although an endogenous ligand for Nr4a1 has not been identified, the naturally occurring small molecule Cytosporone-B (Csn-B) induces activity and subsequent transcription of *Nr4a1* in vivo[38]. Indeed, both acute (6 h) and repeated (8 injections total/2× per day for 4 days; 1-day and 4-days post injection) systemic administration of Csn-B activated *Nr4a1* in the NAc (Fig. 5a, b). Furthermore, acute administration of Csn-B activated *Cartpt* but repeated administration had no effect on *Cartpt* activation at 1-day, but not 4-days following repeated Csn-B administration (Fig. 5c). Acute administration of Csn-B enriched H3K27me3 and H3K4me3, but not H3K27ac at the *Cartpt* promoter (Fig. 5d–f). There were no changes in the enrichment of H3K27me3, H3K4me3 or H3K27ac following repeated Csn-B administration (Fig. 5d–f). Finally, we performed cocaine CPP following repeated injections of Csn-B and found attenuated cocaine-preference (Fig. 5g, h), accompanied by increased *Nr4a1* and *Cartpt* activation, compared to DMSO-injected controls (Fig. 5i). As expected, an acute injection of Csn-B following conditioning also attenuated cocaine-preference and activated *Cartpt* (Fig. 5j–l). There was no difference in total distance traveled between groups, a measure a baseline locomotor activity (Supplementary Fig. 2E–H). Taken together, Csn-B administration was sufficient to activate *Nr4a1* in the NAc, enhance homeostatic gene expression via epigenetic mechanisms and attenuate cocaine CPP.

## Discussion
The majority of studies at the cellular level have focused on cocaine-induced adaptations at either early or late abstinence, which limits understanding of how processes that occur during abstinence affect subsequent behavior. In the present study, we used an unbiased genome wide approach to profile gene expression at both early and late abstinence following cocaine SA. We found that cocaine transiently activated *Nr4a1*, specifically in NAc, while key homeostatic target genes were activated at late abstinence. Interestingly, *Nr4a1* regulates contextual fear memory via transient expression that peaks 2 h after learning but returns to baseline by 4 h, long before memory retrieval at 24 h[39]. It is not yet understood how *Nr4a1* activation patterns cause long-term changes in neuronal function following cocaine exposure. To overcome this, we designed a CRISPR-Cas9 system to activate and repress Nr4a1 expression in the context of cocaine and control conditions. Similar to cocaine induced activation of *Nr4a1*, we found that *Nr4a1* activation bidirectionally regulated Cartpt expression and associated hPTMs. Remarkably, activation of *Nr4a1* attenuated, while repression of *Nr4a1* enhanced behavioral responses to cocaine. To complement these findings, we found that Csn-B, a pharmacological activator of *Nr4a1*, attenuated cocaine behavior. In sum, we identified a mechanism of epigenetic regulation to combat addiction that can be targeted pharmacologically and through locus-specific *Nr4a1* activation.

We hypothesized that *Nr4a1* is a key mediator of homeostatic responses across cocaine abstinence, given that (1) *Nr4a1* is highly expressed in striatal regions of dopaminergic output, such as the NAc (2) *Nr4a1*-deficient mice show increased sensitivity to psychostimulants measured by amphetamine-induced locomotor activity and dopamine metabolism[25,26,40] and (3) following repeated dopamine excitation, the activation of *Nr4a1* prevents the accumulation of spines on dendrites, thereby maintaining normal spine distribution[12,13,28]. Prior data on the regulation of *Nr4a1* and target genes across cocaine abstinence, although scant, supports our findings. For example, one study has applied RNA-sequencing across cocaine abstinence[11], identifying Nr4a1, among other nuclear receptors, as an upstream master regulator of gene expression at late abstinence. An earlier study using microarray, finds that *Nr4a1* is upregulated at 1-day and 10-days of abstinence in rat NAc, following 10 days of cocaine SA (24 h per day)[10]. This abstinence-induced Nr4a1 expression was not detected by qPCR;[41] which may reflect differences in the mRNA quantification method. Importantly, prior studies confirm transient activation of *Nr4a1* by psychostimulant exposure, such that an acute injection of cocaine[42] or methamphetamine[43] upregulates Nr4a1 expression in mouse and rat NAc, respectively.

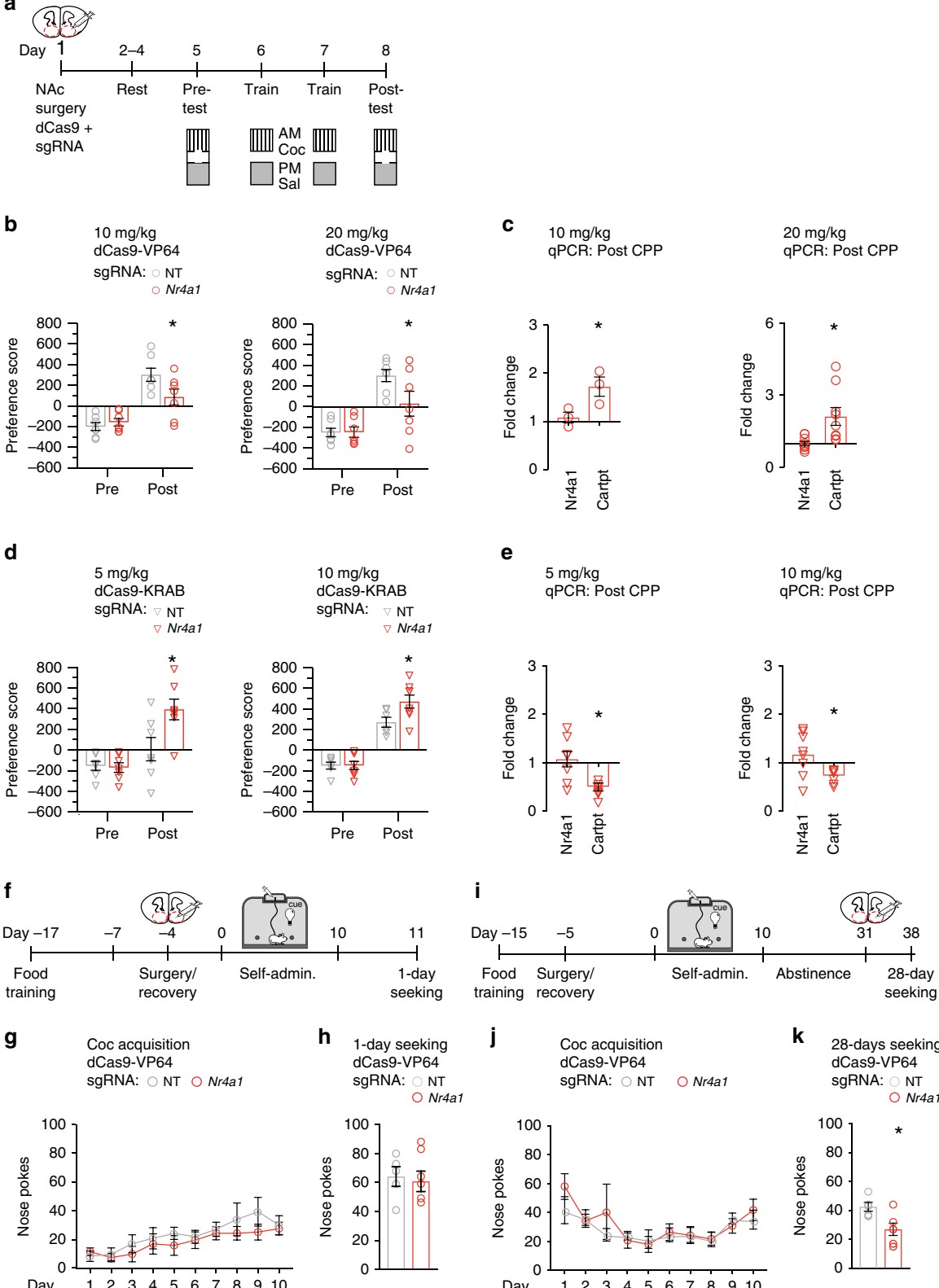

Interestingly, activation of *Nr4a1* following acute methamphetamine treatment is abrogated by prior chronic treatment[43], supporting the concept of long-lasting effects on the *Nr4a1* promoter. Our data expand upon these findings, providing a potential epigenetic mechanism for stimulant activation of *Nr4a1*. Specifically, activation of *Nr4a1* at 1-day of cocaine abstinence was accompanied by enrichment of the activating hPTM, H3K4me3. By

28-days of abstinence, we measured sustained enrichment of H3K27ac and H3K4me3, yet *Nr4a1* was no longer activated above saline controls. Despite this, Nr4a1 remained enriched in the promoter of a key homeostatic target gene, *Cartpt*, in cocaine versus saline treated NAc after 28-days of abstinence.

To examine the functional relevance of *Nr4a1* in the context of cocaine abstinence, we applied CRISPR-mediated gene regulation,

**Fig. 4 CRISPR-mediated Nr4a1 regulation bidirectionally modulates cocaine behavior. a** Schematic of the cocaine conditioned place preference (CPP) paradigm and transfection time course. **b** dCas9-VP64 and sgRNA-Nr4a1 attenuated CPP at two doses of cocaine (10 mg/kg, $n = 7$ mice/group, two-way repeated measures ANOVA, followed by Bonferroni's multiple comparison test, sgRNA-Nr4a1 vs. -NT Pretest $P > 0.9999$, post-test $P = 0.0258$; 20 mg/kg, $n = 7$ mice/group, two-way repeated measures, followed by Bonferroni's multiple comparison test, sgRNA-Nr4a1 vs. -NT pretest $P > 0.9990$, post-test $P = 0.0326$), relative to sgRNA-NT. *$P < 0.05$. **c** dCas9-VP64 and sgRNA-Nr4a1 activated Cartpt following cocaine CPP (10 mg/kg: $n = 3$, unpaired two-tailed t-test, Nr4a1 $P = 0.606$, Cartpt $P = 0.048$; 20 mg/kg: $n = 7$, unpaired two-tailed t-test, Nr4a1 $P = 0.8667$, Cartpt $P = 0.0112$), normalized to GAPDH, and sgRNA-NT. *$P < 0.05$. **d** dCas9-KRAB and sgRNA-Nr4a1 enhanced CPP at two doses of cocaine (5 mg/kg: $n = 7$ mice/group, two-way repeated measures ANOVA, followed by Bonferroni's multiple comparison test, sgRNA-Nr4a1 vs. -NT pretest $P > 0.9999$, post-test $P = 0.0057$; 10 mg/kg: $n = 6$–8 mice/group, two-way repeated measures ANOVA, followed by Bonferroni's multiple comparison test, sgRNA-NT pretest $P > 0.9999$, post-test $P = 0.018$), relative to sgRNA-NT. *$P < 0.05$. **e** dCas9-KRAB and sgRNA-Nr4a1 attenuated Cartpt following cocaine CPP (5 mg/kg: $n = 7$, unpaired two-tailed t-test, Nr4a1 $P = 0.724$, Cartpt $P < 0.0001$; 10 mg/kg: $n = 6$–8, unpaired two-tailed t-test, Nr4a1 $P = 0.4673$, Cartpt $P = 0.046$), normalized to GAPDH, and sgRNA-NT. *$P < 0.05$. **f** Schematic of cocaine self-administration paradigm and transfection time course. **g** dCas9-VP64 and sgRNA-Nr4a1 causes no significant changes in the acquisition of cocaine self-administration, ($n = 5$–6 mice/group, two-way repeated measures ANOVA), relative to sgRNA-NT. *$P < 0.05$. **h** dCas9-VP64 and sgRNA-Nr4a1 causes no significant changes in cocaine-seeking at 1-day of abstinence (unpaired two-tailed t-test, $P = 0.742$), relative to sgRNA-NT. *$P < 0.05$. **i** Schematic of cocaine self-administration paradigm and transfection time course. **j** There were no significant changes in the acquisition of cocaine self-administration, prior to Nr4a1 activation ($n = 6$ mice/group, two-way repeated measures ANOVA), relative to sgRNA-NT. *$P < 0.05$. **k** dCas9-VP64 and sgRNA-Nr4a1 attenuated cue-induced seeking behavior at 28-days of abstinence ($n = 6$, unpaired two-tailed t-test, $P = 0.009$), relative to sgRNA-NT. *$P < 0.05$. All error bars represent mean +/− s.e.m. Source data and statistics provided as a Source Data file.

to drive activation of Nr4a1 to physiologically-relevant levels matching that of cocaine SA. Using this approach, we found that activation and repression of Nr4a1 was sufficient to bidirectionally regulate expression of Cartpt. We then applied this tool to determine epigenetic mechanisms for the delayed activation of Cartpt only at late abstinence. Based on our data, we propose the following mechanism depicted in Fig. 6: In early abstinence, the Cartpt promoter is enriched in activating hPTMs, but its expression is repressed by enrichment of H3K27me3. During late abstinence, this repression is relieved by depletion of H3K27me3 and enrichment of H3K27ac and H3K4me3, via Nr4a1 binding. It is important to consider that the time course of target gene expression following CRISPR activation of Nr4a1 differs from that of cocaine SA. That is, Cartpt is only activated simultaneously with Nr4a1 activation, while in the context of cocaine abstinence, Cartpt is only activated at late abstinence (28-days), when Nr4a1 expression is no longer elevated. This suggests that Nr4a1 is sufficient for activation of Cartpt, but additional factors initially suppress expression during early abstinence. There is evidence the enrichment of H3K27me3 and H3K4me3 recruit nucleosome machinery important in transcription factor binding[44]. In fact, Nr4a1 binding positively correlates with H3K27ac enrichment and creates large scale open chromatin regions facilitating subsequent gene activation via the recruitment of, p300, a histone acetyltransferase[45,46]. We show that Nr4a1 activation depleted H3K27me3 and enriched H3K27ac facilitating activation of Cartpt during abstinence. Moreover, work in peripheral tissue has shown Cartpt activation is regulated by the depletion of H3K27me3 and H3K27ac enrichment, which enables more sensitive activation upon stimulation[47]. These data support our hypothesis that Nr4a1 expression facilitates the cascade of events necessary for activation of repressed genes via H3K27ac enrichment and the resolution of H3K27me3[44].

Prior data on Cartpt supports our supposition of its role in the homeostatic response to cocaine stimulation. First, postmortem brains studies of human cocaine users demonstrate increased CARTPT mRNA[32]. Second, intra-NAc injection of Cart (peptide encoded by Cartpt), only in the presence of psychostimulants, increases dopamine metabolism, and attenuated cocaine locomotor sensitization and SA[24,31,48]. Indeed, intra-NAc injection of Cartpt directly inhibits locomotor activity associated with the intra-Nac injection of dopamine[49]. In this context, Cart functions to reduce levels of dopamine in the synapse, thereby mitigating the effects of dopamine toxicity[24]. We postulate in addition to

Cartpt, Nr4a1 regulates a network of changes which reverse the effects of cocaine in key brain reward regions.

We further applied CRISPR-mediated Nr4a1 activation or repression to the regulation of drug-related behavior. We found that activation of Nr4a1 was sufficient to attenuate cocaine CPP, a phenomenon that was accompanied by increased activation of Cartpt. Alternatively, repression of Nr4a1 increased cocaine CPP, along with repression of Cartpt. Taken together, these findings support our hypothesis that Nr4a1 regulates cocaine reinforced behavior, via regulation of Cartpt, a key homeostatic target gene. Given that we initially identified Nr4a1 in a transcriptomic screen following cocaine SA, we next examined its manipulation in this paradigm[11]. Activation of Nr4a1 during abstinence was sufficient to attenuate cocaine-seeking at 28-days of abstinence, a time point at which Cartpt is normally activated. There are key differences between intraperitoneal (IP) and intravenous (IV) drug exposure paradigms, such as peak levels of cocaine in the brain, which could explain discrepancies in our findings[50,51]. Specifically, the concentration of cocaine in the brain increases following repeated IP administration but remains constant following repeated IV administration[50,51]. Because we measured the same effect on Nr4a1 activation in both paradigms, we provide evidence that the mechanism of Nr4a1 activation is conserved between the two paradigms. Nr4a1 may regulate gene expression essential in long-term context-dependent drug associations rather than changes directly associated with the pharmacological properties of cocaine. This notion is supported by the lack of effect of Nr4a1 activation on cocaine SA acquisition. The regulatory effects of Nr4a1 on cocaine-induced gene transcription and its role in attenuating cocaine-induced behaviors make it a promising therapeutic target in cocaine addiction.

Indeed, our results indicate the preclinical efficacy of Csn-B, a small-molecule agonist of Nr4a1, in activating Nr4a1 in NAc, attenuating cocaine behavioral responses, and enhancing homeostatic gene transcription. We found that Cartpt activation is associated with increased enrichment of H3K27me3 and H3K4me3 following acute of administration of Csn-B. In contrast, we found no changes in Cartpt activation or hPTMs following repeated Csn-B treatment, which suggests that changes in H3K27me3 and H3K4me3 after acute administration are early events indirectly involved in gene activation. It is important to note that we have not tested the effect of Csn-B on delayed activation of Cartpt at later time points which may explain the absence of changes in gene expression and hPTMs following

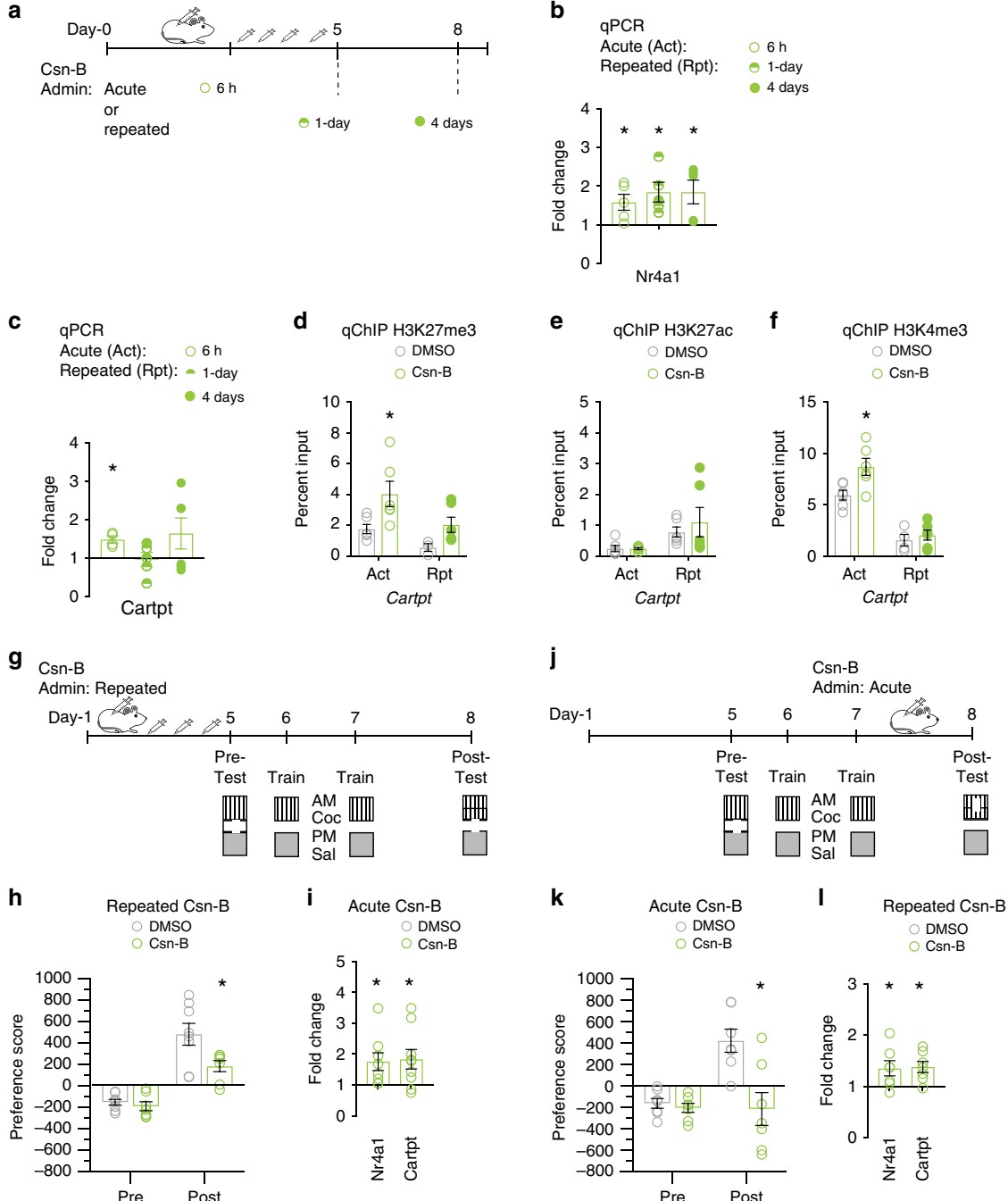

**Fig. 5 Csn-B activated *Nr4a1* and attenuated cocaine behavior. a** Timeline of Csn-B administration (10 mg/kg) acute and repeated (2× daily, 4 days). **b** *Nr4a1* was activated after acute and repeated Csn-B treatment ($n = 5$ mice/group; two-way ANOVA, followed by Bonferroni's multiple comparisons test, 6 h $P = 0.028$, 1-day $P = 0.0174$, 4-days $P = 0.0168$), normalized to GAPDH, relative to DMSO controls. *$P < 0.05$. **c** Csn-B administration activated *Cartpt* after acute but not following repeated Csn-B administration (two-way ANOVA, followed by Tukey's multiple comparisons test, 6 h $P = 0.0307$, 1-day $P > 0.9999$; 4-day $P = 0.0978$), normalized to GAPDH, relative to DMSO controls. *$P < 0.05$. **d** Acute Csn-B administration enriched H3K27me3 at the *Cartpt* promoter ($n = 6$ mice/group; two-way ANOVA, followed by Tukey's multiple comparisons test, 6 h $P = 0.0276$), normalized to input, relative to DMSO controls. *$P < 0.05$. **e** There were no significant change in H3K27ac enrichment following acute and repeated Csn-B administration ($n = 6$ mice/group; two-way ANOVA, followed by Tukey's multiple comparisons test, 6 h $P = 0.0276$), normalized to input, relative to DMSO controls. *$P < 0.05$. **f** Acute Csn-B administration enriched H3K4me3 at the *Cartpt* promoter ($n = 6$ mice/group; two-way ANOVA, followed by Tukey's multiple comparisons test, 6 h $P = 0.033$), normalized to input, relative to DMSO controls. *$P < 0.05$. **g** Schematic of cocaine conditioned place preference (CPP) paradigm and Csn-B administration time course. **h** Repeated Csn-B administration (green circles) attenuated cocaine CPP ($n = 7$ mice/group, two-way repeated measures ANOVA), relative to DMSO control. *$P < 0.05$. **i** Repeated Csn-B administration activated *Nr4a1* and *Cartpt* following cocaine CPP ($n = 7$-8, unpaired two-tailed *t*-test, *Nr4a1* $P = 0.0157$, *Cartpt* $P = 0.005$), normalized to GAPDH, relative to DMSO. *$P < 0.05$. **j** Schematic of cocaine conditioned place preference (CPP) paradigm and Csn-B administration time course. **k** Acute Csn-B administration inhibited cocaine CPP ($n = 7$ mice/group, two-way repeated measures ANOVA) relative to DMSO control. *$P < 0.05$. **l** Acute Csn-B administration activated *Nr4a1* and *Cartpt* following cocaine CPP ($n = 7$, unpaired two-tailed *t*-test, *Nr4a1* $P = 0.0404$, *Cartpt* $P = 0.0075$), normalized to GAPDH, relative to DMSO. *$P < 0.05$. All error bars represent mean $+/-$ s.e.m. Source data and statistics provided as a Source Data file.

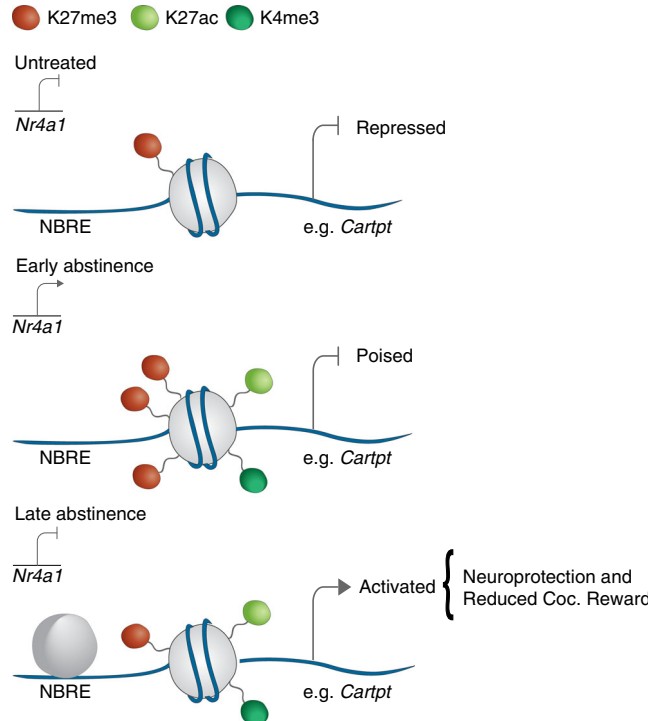

**Fig. 6 Mechanism of *Nr4a1* activation of homeostatic gene expression.** In untreated mice, *Cartpt* is repressed. During early abstinence, *Nr4a1* is activated and *Cartpt* is repressed by H3K27me3 enrichment but poised for later activation. During late abstinence, Nr4a1 expression returns to baseline, and *Cartpt* is activated by depletion of H3K27me3 and enrichment of H3K27ac and H3K4me3 via sustained Nr4a1 binding.

repeated Csn-B administration. Despite these inconsistencies, we found that both acute and repeated systemic administration of Csn-B attenuated cocaine CPP and activated *Cartpt*. This psychostimulant specific effect suggests cocaine and *Nr4a1* induce synergistic effects on cocaine activation of *Cartpt*.

Cue-associated relapse is a key symptom of addiction that increases in likelihood across abstinence. It is promising, however, that drug-induced synaptic plasticity and associated drug memories are reversible, as evidenced by the fact that human addicts show increased reactivity to drug-cues for up to 6 months but not at 1 year of abstinence[8]. A critical mechanism in the formation of stable drug-associated memories is the persistence of changes in synaptic and structural plasticity in NAc, measured by a stable elevation in glutamate receptors and spine density, seen at 30 days of abstinence but not 1-day[7,52,53]. Such changes in plasticity and behavior dissipate by 180 days which suggests there are homeostatic mechanisms working to restore normal brain function[7]. Prior studies support the notion that gene expression in NAc underlies neuronal activity mediating the incubation of cocaine craving[1,54]. Our work identifies a molecular mechanism that reverses drug-induced pathology and is a potential inroad to therapeutic interventions in addiction. Given the fact that most drug addicts are poly drugs users, targeting a common mechanism that generalizes to other drugs is essential. While future experiments are needed to elucidate the long-term effects of this treatment on other drugs, and its role in attenuating cocaine SA, our current findings underscore the value in pursuing research on the mechanism of action of *Nr4a1* in relapse behavior and addiction in general.

In conclusion, our data support the role of *Nr4a1* in activating *Cartpt* and suppressing cocaine-induced behavior in late abstinence. Locus-specific gene regulation allowed elucidation of the precise epigenetic mechanisms involved in *Nr4a1* target gene

activation, specifically the loss of repressive and gain of activating histone modifications at late abstinence. Finally, the availability of a small molecule agonist that crosses the blood-brain barrier to activate *Nr4a1*, in vivo, lends promise to the clinical relevance of our findings, and the potential of *Nr4a1* as a therapeutic target to combat cocaine addiction.

## Methods

**Animals**. Adult male and female, 8-week-old C57BL/6J mice (The Jackson Laboratory) were used in this study. Mice were housed on a 12-h light-dark cycle at constant temperature (23 °C) with access to food and water ad libitum. Animals were habituated to experimenter handling for at least 1 week before experimentation. All animals were maintained according to National Institutes of Health guidelines in Association for Assessment and Accreditation of Laboratory Animal Care. Ethical and experimental considerations were approved by the Institutional Animal Care and Use Committee of The University of Pennsylvania.

**Cocaine intravenous self-administration**. Mice were single housed and habituated to the researcher by daily handling sessions for 7 days prior to the start of SA training. Operant responding was defined as active if paired with the delivery of a reinforcer, and inactive if paired with no reinforcer or vehicle. Active responding resulted in the presentation of a reinforcer, paired with light and tone. All studies were conducted on a fixed-ratio 1 (FR1). The acquisition of the instrumental task (nose poke or wheel spin) was facilitated in naïve mice by 10 daily, 1-h training sessions for operant responding to receive a food pellet reinforcer (Bio-Serv, Product #F0071; 20 mg). Mice were introduced to the food pellets and food restricted for 3 days prior to the start of the operant training.

Following food SA, mice were implanted with an indwelling catheter to the right external jugular vein under ketamine (80 mg/kg) and xylazine (12 mg/kg) anesthesia, followed by 3 days of recovery, during which time they were monitored daily for distress. Mice were then assigned to self-administer either saline or cocaine (0.7 mg/kg/infusion) on an FR1 schedule during daily 2-h sessions for 21 consecutive days. We defined acquisition as the first of three consecutive sessions during which a mice consumed an average of 5 mg/kg cocaine, or about ten infusions (5.6–8.4 mg/kg), infusions did not vary by more than 20%, and at least 80% of responses were on the active wheel. Mice were sacrificed 1-day and 28-days after the last cocaine session, brains were removed and NAc, VTA, and PFC was dissected using 2 mm aluminum Harris micro-punch (Sigma-Aldrich). Tissue was frozen on dry ice and stored at −80 °C for downstream analysis.

Cocaine-seeking experiments: Following 10 days of cocaine SA mice underwent 1-day or 28 days of abstinence. On day 21 of abstinence, NAc was targeted by stereotaxic injection, as described below. For seeking tests, mice were subjected to a single 2-h SA session, in which no reinforcer was given for active responses, but all cues were present. Mice were sacrificed immediately following the seeking session and brain region(s) collected for downstream analysis.

**Investigator administered cocaine**. Mice were group housed and habituated to the researcher by daily handling sessions for 3 days. Mice received 10 daily injections of cocaine hydrochloride dissolved in 0.9% saline (i.p., 20 mg/kg) or 0.9% saline in clean cages and allowed to move freely for 30 min, before being returned to home cages. NAc tissue was collected 1-day and 28 days after the last injection.

**Conditioned place preference**. Mice were placed in a two-sided conditioning chamber (Ugo Basile) for 20 min to assess innate pretest preferences. Mice were then conditioned for 2-days to associate cocaine with their non-preferred side. Mice were injected in the morning with cocaine and in the afternoon with saline (i.p., 5, 10, or 20 mg/kg), then restricted to their preferred chambers and non-preferred, respectively, for 30 min. On day 4, mice were placed in the chamber with free access to both sides for a 20 min post-test session to assess conditioned place preference. ANY-Maze (version 4.99) software was used to analyze time and total distance traveled. NAc tissue was collected 4 h following the post-test session. Cocaine preference for all testing sessions was calculated as time spent in the cocaine-paired chamber minus time spent in the saline-paired chamber and is reported as a preference score (in seconds). Mice were defined as having acquired cocaine CPP when the average cocaine preference score of the cocaine-treated group was significantly higher than the preference score at baseline[55].

**Intra-NAc transfection**. NAc was targeted bilaterally[56,57] using the following stereotaxic coordinates: +1.6 (anterior/posterior), +1.5 (medial/lateral), and −4.4 (dorsal/ventral) at an angle of 10° from the midline (relative to Bregma). In-vivo transfection were conducted using the transfection reagent Jet-PEI (Polyplus transfection), prepared according to manufacturer's instructions. 12.5 μL DNA plasmid (1.0 μg/μL) was diluted in 12.5 μL of 10% sterile glucose and added to diluted Jet-PEI, mixed by pipetting and incubated at room temperature (RT) for 15 min. A total of 1.5 μL of Jet-PEI/Plasmid solution was delivered NAc at a rate of 0.2 μL per minute, followed by 5 min of rest. Following surgery, mice were allowed to recover for 4–7 days and closely monitored for distress

**Cytosporone B treatment**. Mice were group housed and habituated to the researcher for 3-days by daily handling sessions. Cytosporone B (Sigma-Aldrich) reconstituted with DMSO was freshly dissolved in 0.9% saline and injected i.p. (10 mg/kg)[38,58]. Mice were either treated with single or repeated injections (twice-daily injections for 4 days). Control mice were injected with DMSO diluted in saline (0.9% saline). The final concentration of DMSO was no more than 0.1%.

**Tissue collection, RNA extraction, and qRT-PCR**. For tissue collection, mice were euthanized by cervical dislocation and the brain was placed in ice-cold phosphate buffered saline with protease inhibitor cocktail (Roche cOmplete Protease inhibitor). One millimeter thick sections of prefrontal cortex, nucleus accumbens and ventral tegmental area, brain regions were prepared using a cold tissue matrix and regions were micro-dissected with a 2 or 1.2-mm diameter micro-punch (Harris) under a fluorescence stereoscope (Leica). For in-vivo transfection studies, transfected tissue was visually inspected for confirmation of NAc targeting and dissection. RNA was isolated using the RNeasy Mini Kit (Qiagen) according to the manufacturer instructions. Primer sequences can be found in Supplementary Table 1. Data was analyzed by comparing Ct values of the experimental group to control using the ΔΔCt method.

**RNA-seq data analysis**. RNA-seq reads were aligned against mouse reference genome (mm9, ensemble annotation) using STAR (Version 2.4.1d)[59] with default parameters. Aligned reads were normalized using different methods including TMM[60], TMM_CPM[60], RPKM[61], EDAseq_UQ[62], FullQuantile[62], Median, PoissonSeq[63], RUV[64], and PORT (https://github.com/itmat/Normalization). Differential gene expression analysis was performed using different models: DEseq2-Wald[65], EdgeR-Robust-LRT[61], LimmaVoom[66], Mann-Whitney[67] and Parametric t-test[68]. Q-values are determined for each of the 45 combinations of normalization and differential gene expression analysis. Genes with a mean q value smaller than 0.25 were considered a target of differential expression for each combination. A gene is identified as differentially expressed gene if it is detected by the combination of five different methods.

**Venn diagrams, volcano plot, and heatmaps**. R packages Venn diagram and heatmaps were used to generate Venn Diagram and heatmap figures. Volcano plots were produced using R package ggplot2. Gene expression level in the heatmap and volcano plots were calculated based on RPKM normalization using "rpkm" function from edgeR package. Gene length was calculated based on the length information from ensemble GFF annotation file. Genes with multiple isoforms were collapsed and the length was calculated according to the longest collapsed isoform.

**Chromatin immunoprecipitation**. ChIP was performed on bilateral NAc 2 mm punches dissected as described above from 1 mouse, 24 h after the last drug treatment or abstinence[69]. For in vivo CRISPR studies unilateral NAc 1.2 mm punches were pooled from 5 to 6 mice. Chromatin was sheared using a diogenode bioruptor XL at high sonication intensity. For histone modification enrichment, chromatin was sheered for 30 min (30 s on/30 s off) and fragment size was verified at 150–300 bps with an Agilent 2100 bioanalyzer. For Nr4a1 binding enrichment, chromatin was sheered for 22 min (30 s on/30 s off) fragment size was verified at 400–600 bps. Sheared chromatin was incubated overnight with the following antibodies previously bound to magnetic beads (Dynabeads M-280, Life Technologies): antibody to H3K4me3 (EMD Millipore 07-473), antibody to H3K27me3 (EMD Millipore 07-449), antibody to H3K27ac (EMD Millipore 07-360), antibody to Nr4a1 (Novus NB100-56745), IgG (Novus NBP2-24891). The Dynabeads were washed twice each with 1 ml of Low Salt Wash Buffer (20 mM Tris, pH 8.0, 150 mM NaCl, 2 mM EDTA, 1% TritonX-100, 0.1% SDS), High Salt Wash Buffer (20 mM Tris, pH 8.0, 500 mM NaCl, 2 mM EDTA, 1% TritonX-100, 0.1% SDS), and TE Buffer (10 mM Tris, pH 8.0, 1 mM EDTA). After reverse cross-linking and DNA purification (Qiagen Spin Column), primers were designed to amplify regions −100 to +100 bps spanning the Nr4a1-binding motif within gene promoter regions. qChIP primer sequences are be found in Supplementary Table 2.

**Neuro2a transfections**. Neuro2a (N2a) cells ($n = 1 \times 10^6$ cells) [CCL-131, ATCC (manufacturer authentication available on-line)] were cultured and transfected with a total of 300 ng of plasmid DNA using Effectene reagent (Qiagen)[57]. RNA was isolated using the RNeasy Mini Kit (Qiagen) according to the manufacturer instructions. qPCR and data analysis were performed as described above.

**Engineered DNA-binding molecule ChIP (enChIP)**. enChIP (Active Motif) was performed according to the manufacturer's instructions with the following modifications[36]. N2a cells ($n = 10 \times 10^6$ cells) were transfected (Effectene) with 1 μg of pAM_dCas9 (Active Motif) and 1 μg of sgRNA-231. Cells were fixed 48 h post-transfection with 1% formaldehyde at 37 °C for 5 min. The chromatin fraction was extracted, fragmented by sonication and 150–300 bp fragments were confirmed with the Agilent Bioanalyzer. The sonicated chromatin was pre-cleared with 15 μg of normal mouse IgG (Active Motif) conjugated to 150 μL of Dynabeads-Protein G (Invitrogen) and subsequently incubated with 15 μg of AM-Tag polyclonal antibody (Active Motif 61677), previously conjugated to 150 μL of Dynabeads-Protein

G. The Dynabeads were washed twice each with 1 mL of Low Salt Wash Buffer (20 mM Tris, pH 8.0, 150 mM NaCl, 2 mM EDTA, 1% TritonX-100, 0.1% SDS), High Salt Wash Buffer (20 mM Tris, pH 8.0, 500 mM NaCl, 2 mM EDTA, 1% TritonX-100, 0.1% SDS), and TE Buffer (10 mM Tris, pH 8.0, 1 mM EDTA). The Dynabeads were suspended in 285 μL of TE and 12 μL of 5 M NaCl and incubated at 65 °C overnight for reverse crosslink. After RNase A treatment at 37 °C for 1 h, samples were treated with Proteinase K at 65 °C for 2 h. Subsequently, DNA was reverse crosslinked and purified (Qiagen Spin Column). qChIP primers were designed to amplify regions between −100 and −350 bps spanning the sgRNA-231 targeting sequence within the Nr4a1 gene promoter and a location 1 kb upstream. Primer sequences are found in Table 2.

**Immunohistochemistry**. Mice were unilaterally transfected with dCas9-VP64 or transfection reagent alone (mock) and sacrificed at 1-day post transfection. Animals were then transcardially perfused with ice-cold DPBS, followed by ice-cold 4% paraformaldehyde (PFA). Brains were fixed overnight in 4% PFA at 4 °C, and then cryoprotected in 30% sucrose solution overnight at 4 °C. Coronal brain sections (45 μm) were collected using a sliding microtome (Leica, SM2010R), and serial sections were stored at −20 °C in 96-well plates containing anti-freeze solution (300 g sucrose, 300 mL ethylene glycol, 500 mL 0.1 M PBS). Brain sections were washed in TBS with 0.05% TritonX-100 and then incubated in primary antibody solution (3.33% donkey serum and 0.05% TritonX-100 in TBS) with anti-Cas9 (Catalog # 61578, Active Motif, 1:500) overnight at 4 °C. Brain sections were washed in TBS with 0.05% TritonX-100 and then incubated in secondary antibody solution (3.33% donkey serum and 0.05% TritonX-100 in TBS) with Alexa Fluor 555 (1:250) and DAPI nuclear stain (Roche, 1:1000) for 1–2 h. at room temperature. Brain sections were washed in TBS with 0.05% TritonX-100 and then incubated in a second primary antibody solution with anti-NeuN, AlexFluor 488 conjugated (Catalog # MAB377X, Millipore, 1:500) overnight at 4 °C. Brain sections were washed in TBS with 0.05% TritonX-100 and then were mounted with 2.5% PVA/DABCO mounting media (Sigma). Brain sections were imaged using as z-stacks using the 40× objective of a Zeiss LSM 810 confocal microscope (Carl Zeiss).

**Statistics**. The appropriate statistical test was determined based on the number of comparisons being done. Student's t-tests were used for comparison of two groups, in the analysis of qRT-PCR following in vivo CRISPR-mediated gene activation and Two-way ANOVAs were used for comparison of saline and cocaine treatment across abstinence time points; when appropriate, a post-hoc test followed to determine significant differences across multiple comparisons, in the analysis of qRT-PCR and qChIP data. One-way ANOVA was used for analysis of three or more experimental groups, when appropriate, a post-hoc test followed to determine significant differences across multiple comparisons, in the analysis of CRISPR-mediated gene regulation when compared to controls (qRT-PCR). Repeated measure two-way ANOVA was used for comparison of two groups on different observations, when appropriate, a post-hoc test followed to determine significant differences across multiple comparisons, in the analysis of cocaine self-administration and condition place preference. Main and interaction effects were considered significant at $P < 0.05$. P-values greater than 0.05 and below 0.1 were considered trends. Data are expressed as mean [+ or −] s.e.m. The Grubbs test was used when appropriate to identify outliers. F-tests of variance were conducted on all data sets to ensure that the data followed a normal distribution. All experiments were carried out one to three times, and data replication was observed in instances of repeated experiments. Experimental sample sizes were determined using G*power using preliminary data. Details on each statistical test can be found in the source data file.

**Reporting summary**. Further information on research design is available in the Nature Research Reporting Summary linked to this article.

## Data availability

RNA-Seq source data presented in Figs. 1, 2 and Supplementary Figs. 2, 4 can be accessed through GSE141520. All data and statistics reported in main text can be found in the Source Data file.

## Code availability

ANY-Maze software was used for behavioral tracking in conditioned place preference studies. We have used published software for data analyses. Requests can be sent to the corresponding author.

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

## Acknowledgements

We are grateful to Dr. Marisa S. Bartolomei and Dr. Montserrat C. Anguera for critical reading of this manuscript, and to members of the Perelman School of Medicine Next Generation Sequencing Core, including Dr. Jonathan Schug, Olga Smirnova and Shilpa Rao, for their contributions. Financial support is kindly acknowledged from Charles E. Kaufman Foundation Young Investigator Award (E.A.H.), Whitehall Foundation Grant (E.A.H.), NIH-NIDA Avenir Director's Pioneer Award (E.A.H., DP1 DA044250) and Research Supplements to Promote Diversity in Health-Related Research (E.A.H., M.D.C., DP1 DA044250-01), T32 Predoctoral Training Grant in Pharmacology (M.D.C., T32GM008076), NIDA Research Project Grant (R.C.P. NIDA R01 DA33641, NIDA R01 DA15214), NIDA K01 Mentored Research Scientist Career Development Award (M.E.W. DA039038). We would also like to thank the NIDA drug supply program for providing the drugs used within this study.

## Author contributions

M.D.C., M.E.W., R.C.P., and E.A.H. designed and analyzed behavioral experiments, performed primarily by M.D.C., with the assistance of S.I.L. and K.S.C. M.D.C., Q.H., and E.A.H. designed and analyzed RNA-SEQ experiments, performed by Q.H. M.D.C., and E.A.H. designed and analyzed qPCR, qChIP, in vivo and in vitro CRISPR studies, performed by M.D.C. M.D.C., A.M.B., H.S., and E.A.H. designed and analyzed IHC experiments, performed by A.M.B. C.J.L., M.D.C., and E.A.H. designed schematics performed by C.J.L. M.D.C., and E.A.H. wrote and revised the manuscript.

## Competing interests

The authors declare no competing interests.
