## [Peer Review File · Nature Communications]

Reviewers' comments:

Reviewer #1 (Remarks to the Author):

This article is attempting to identify potential homeostatic mechanisms in the NAc that are working to reverse drug-induced pathology after cocaine exposure, thereby reducing risk of relapse. They interestingly found that cocaine self-administration transiently activates Nr4a1 in the NAc, subsequently contributing to activation of homeostatic genes during late abstinence. Previously Nr4a1 has been identified as an upstream regulator of gene expression during late abstinence, so this is not the first work to suggest an important role of Nr4a1 in regulating gene expression following cocaine exposure. This paper built on these previous findings with genome wide RNA sequencing and by identifying epigenetic mechanisms for the activation of Nr4a1, namely changes in histone modifications. Furthermore, although Nr4a1 was only enriched after 1 day of abstinence, it was still enriched in the promoters of homeostatic genes such as *cartpt* and *vmat2* at 28 days. The authors then drove Nr4a1 activation to physiologically-relevant levels (matching cocaine IVSA), and find that activation and repression of *nr4a1* increased and decreased expression of *cartpt* and *vmat2* respectively. The expression of *cartpt* and *vmat2* when *nr4a1* is active (ie with viral transfection or after cocaine IVSA) might be dependent on the enrichment of repressive hPTMs. This *nr4a1* induced expression of *cartpt* and *vmat2* could be functionally relevant to decreased cocaine-related behaviors as *nr4a1* over-expression, both virally and pharmacologically, decreased cocaine conditioned place preference and viral over-expression reduced cocaine seeking. These results are very compelling and suggest that these homeostatic genes, *cartpt* and *vmat2*, could be potential therapeutic targets for addiction, particularly considering there is a potential pharmacological treatment. However, there are some issues with the language and statistics used and a few additional experiments seem necessary to complete this interesting story.

Major:

Statistical issues:

There are several statistical issues in this manuscript. The major issue is the use of t-tests throughout instead of ANOVAs since day can be used as a factor. T-tests after 1 or 28 days of abstinence are used for each gene of interest or hPTM when it would be more appropriate to use ANOVAs. In some instances the failure to use ANOVAs seems to be favoring t-tests since ANOVAs would result in main effects of cocaine and no effect of # of days. The fact that each group is analyzed by t-test is just not appropriate.

Also, it is hard to tell whether the t-tests used are comparing the drug or virus treatment to the control group or to 1. The control groups (for example Fig3H) are not shown, but the fold change indicates the control groups are at 1. However, it is unclear whether the analysis is being done against 1 versus treatment or the controls, which although set to 1 would maintain variability. This should be made more clear since control data are not shown. Also, figure captions incorrectly say control is shown in grey.

A statistics section also needs to be added to the methods of this paper. A lot of these issues would be made clear if the analyses were laid out in one place. That p-values used should be made more explicit. Especially cut-offs used for RNAseq and what constitutes a trend. For RNAseq the p-value appears to be 0.05, which doesn't seem to be stringent enough. A table should be included with the # of differentially expressed genes across stringencies. Furthermore, only one trending effect was presented in the figure and I found the p-value in the figure as .159. What a "trending" value is should be laid out and if the authors include trends that should be clear in the results, what is significant and what is a trend. Furthermore, any other values that fall within that trending p-value should be called a trend as well.

The authors should mention what post-hoc tests are being performed after significant interactions and whether multiple comparisons are being corrected for.

Conditioned place preference issues:

There are some more major issues with the way the CPP findings are described. I don't disagree that nr4a1 over-expression decreased cocaine preference and repression increases preference, but some things are mentioned that aren't necessarily true. The authors say that the groups spent equal time in both chambers during the pre-test, but that doesn't appear to be true since 40% was spent in the future cocaine-paired side and that indicates a bias. They don't have different biases, which may be the authors point, but this should be made more clear.

The authors also say that both groups spent more time in the cocaine-paired chamber, but that doesn't appear to be true and statistics do not show this. If you draw a line at 50% which is no preference for cocaine that is where nr4a1 over-expression is, indicating mice do not spend more time in cocaine than saline-paired chambers. The authors can either change the text to refer to more time post-test compared to pre-test in the cocaine-paired chamber or you can show the data as post-test – pre-test time in cocaine and just indicate textually that the mice still have a preference for the cocaine paired side compared to the pre-test day.

It is also critically important to show CPP results at multiple doses. A 10 mg/kg dose for is used for the activation experiment and 5 mg/kg for the repression. The authors call the latter dose a sub-threshold dose, but in our, and other labs experience, this is not the case. Mice still condition to much lower doses of cocaine, just to lesser amounts. If analysis is done by seconds instead of percentages that might help depict the data better. The authors should seriously consider running CPP with at least one other dose. It is important to show that the mice aren't impaired in their context associated learning (in fig4b) so having a dose where the mice do condition well is important. Also, it would be nice to see a dose where the controls show preference and whether preference is further increased after nr4a1 repression or is equal to controls.

More issues:

Another major issue seems to generally be why vmat2 is such a large part of this paper. There are many effects that are inconsistent between cartpt and vmat2, some of which suggest that vmat2 could not be contributing to mitigating the behavioral effects seen with activating/repressing nr4a1 etc. One example of this is figure 5b where Csn-B treatment acutely or repeated for 1 day actually decreases vmat2 expression. Also, the authors report a trend for cocaine IVSA to increase vmat2 in RNAseq, but that p-value is a .159. That is much higher than the typical trending p-value of up to 0.1. It seems as though the authors might have found this in their qPCR before RNAseq and were trying to include it in the paper even though the results with vmat2 aren't compelling.

The model the authors propose is also a bit confusing, mostly because of the inconsistent results between cartpt and vmat2, but vmat2 is also included on the figure. The authors have included the presence of 27me3 and 4me3 at day 1, which is fine, and then 4me3 and k27ac at 28 days, but the results actually show that is enriched at 1 day for cartpt as well (it also looks close for vmat2). The bigger issue is with me3, since it is only elevated for cartpt (but again at both times not just 28 as in the figure) but not vmat2, so these two genes appear to be regulated differently. Perhaps the authors can emphasize more the change in repression, where there is not only more repression at day1, but less repression at day 28 with some varying things happening to antinational modifications.

Minor:

Throughout, there are several typos and incorrect grammar. The document should be proofread for these errors.

The authors mistakenly call the cocaine and pellet reinforcers rewards, this language should be

changed.

The authors also say that CPP and ivsa are isomorphic, but they are not. They measure fundamentally different behaviors and many studies have found opposing effects of some manipulation on CPP and IVSA (see for example Nestler lab work with CREB). In fact, the vary paper the authors cited says "CPP measures a learning process that is fundamentally distinct form drug self-administration". Since both behaviors are present in this paper this is only a textual issue, which should be corrected.

During cocaine self-administration the number of infusions required for criteria was only 10, but the mice seem to be taking much more and the saline mice seem to be taking at least 10. Why is 10 the criteria?

Also in the self-administration studies with the viruses # of infusions and discrimination index are not shown. Since the authors specifically discuss reasons why over-expressing nr4a1 prior to self-administration do not affect behavior, seeing those other measures is important.

A figure should be included in the supplement showing that to be 1 day and to be 28 day abstinent groups do not differ in their cocaine self-administration.

In figure 4, reinstatement should include the inactive nose poking as well. There could be a different interpretation if inactive nose pokes increase when active nose pokes decrease in the nr4a1 over-expressed group.

Throughout the figures many axes do not match between 1 and 28 days. I understand that is to see better, but it can make interpreting the findings to be very difficult. Perhaps the authors can use the same axis and use a hash to symbolize skipping space on the y axes so we can see the data compared or use the same larger axis and then include a zoomed in version as an inset figure.

Figure 3 should include a zoomed out image with the virus as well, not just DAPI to show the expression with the morphology intact to make placement clear. Also the closer image provided shows it is from the entire NAc, but this image isn't big enough for that. The position in the section should be more clear, perhaps by including where the commissure is located. Also, colors should be written on the image so it is clear, which color is which protein. There should also be a scale bar included. None of this information is located in the caption either. This image also doesn't show the NeuN staining very well, a better image should be used where red staining (NeuN?) without colocalization can be readily seen.

I believe the key in 5b is wrong.

The authors should consider running qPCR after the acute Csn-B treatment in figure 5 as well since the behavioral effect was actually larger in that experiment. Also, the authors do not discuss why Vmat2 is not changed after treatment, but is increased after treatment during CPP.

The work with the Csn-B compound is very interesting since it activates nr4a1 but also causes these repressive modifications. However, it also still increases expression of cartpt and vmat2. The authors should discuss how repressive hPTMs would keep nr4a1 from increasing expression for cartpt and vmat2 until day 28 if they are expressed with this compound and so is the repressive modification.

Reviewer #2 (Remarks to the Author):

This is a very interesting and well written animal model study on the role of Nr4a1 transcription factor in cocaine-related reward behavior in the ventral striatum, with key experiments involving in vivo epigenomic editing and Nr4a1 target gene chromatin profiling.

I have very little to comment, other than Figure 2A, supposedly showing NRBE motif enrichment (for Nr4a1 binding) at their candidate gene promoters. It is not clear how the authors arrived at that conclusion, I assume they used sonication to prepare their chromatin so the direct binding of Nr4a1 to a specific motif could not be derived from such type of approach. This issue should be better explained in a revision, or if inconclusive, Figure 2A removed from the paper

Reviewer #3 (Remarks to the Author):

The article by Carpenter et al. examines the role of Nr4a1 in regulating gene expression, histone modifications, and cocaine CPP and self-administration behaviors. The authors first examined both 1d vs 28d of abstinence following either self-administration or experimenter-administered injections of cocaine to model incubation of cue-induced drug craving and found differences in both Nr4a1 gene expression and histone modifications. They examined possible downstream regulation of several Nr4a1 target genes, including *Cartpt* and *Vmat2*, at 1 vs 28d abstinence timepoints. They found that upregulation of *Cartpt* and *Vmat2* gene expression associated with depletion of repressive epigenetic marks, H3K27me3. Using a modified Crispr/Cas system or an agonist to regulate Nr4a1 expression and function, the authors show an impact on contingent and non-contingent cocaine behavior. Overall, this is an interesting study with novel results that demonstrate a role for Nr4a1 in cocaine behavior. However, there are numerous weaknesses that need to be addressed.

Detailed comments:

1. Several results are not well supported due to low N's and inconsistent reporting of statistics. Fig. 1J is unconvincing as a negative result given the variability and small Ns. Here the p-value=0.1193 and is reported as negative, but elsewhere in Figure 2B, a p-value = 0.153 is reported with a "#" while in Figure 3H, a p=0.078 is not reported as a statistical trend. Perhaps these are typos (for instance, another example in the legend of 5D a "0.587" is surely supposed to be a "0.0587" as suggested by the "#" in the figure itself). Nonetheless, clarification and increased power in Figure 2 would help to avoid possible false negatives. Similarly, in Fig. 1K, there is a big increase in the mean of enrichment H3K27me3 on 1 day, although there is no significant difference (p=0.0997, but with only n=3). Authors conclude that "We found no difference in repressive modification, H3K27me3". These experiments are underpowered and therefore the conclusions are unconvincing.
2. The statistical analysis methods are not well described. Please describe the statistical analysis methods in detail. Authors used unpaired two-tailed t test repeatedly in the comparison of more than 3 groups such as Fig 2A, and Fig 3B. It will increase the risk to detect false positive. Please use the appropriate correction for the statistical analyses.
3. Authors concluded that "These findings suggest that abstinence-induced expression of *Cartpt* and *Vmat2* is regulated by sustained Nr4a1 enrichment and depletion of repressive H3K27me3". The depletion of H3K27me3 is a possible mechanism of reduction of *Cartpt* and *Vmat2* expression; however, the causal role of sustained Nr4a1 was not examined. The hPTM changes produced by Nr4a1 are only correlative, and do not consistently explain the changes observed (e.g. locus-specific Nr4a1 manipulations vs. agonist on *Cartpt* and *Vmat2* expression).
4. Not all of the data support the hPTM conclusions in Figure 6, for example, H3K27ac is increased at 1 day in Fig 2G for *Cartpt*, but not *Vmat2* (though this is another odd statistic as there is p=.132, but Ns of only 3 per group). Considering that the qChIP data in Fig 2E shows a significant

effect of Nr4a1 on Cartpt, but not Vmat2, Figure 6 is too simplistic (see also Fig 2H for Vmat2 differences). Perhaps these inconsistencies suggest that Cartpt, but not Vmat2, is important for the behavioral results as both the Cas9 and Csn-B changes are consistent for this locus, but are dissociated from the Vmat2 one?

5. Despite the attempt to use timepoints associated with incubation of drug craving (1 vs 28d abstinence), there is no obvious incubation in the 28d group of mice in Figure 4K. Figure 4H has nose pokes at 1d seeking of about 60, but in a separate group, the nose pokes at 28d seeking are only about 40 for controls. This draws into question the use of these mice in previous figures to study the effects of incubated craving. Why do they not incubate?

6. An acute dose of Csn-b attenuates CPP (Figure 5I), but it does not increase both Cartpt and Vmat2 levels (Figure 5B). Although Nr4a1 clearly has a role, the mechanism suggested to explain its role is unconvincing at present. Does Csn-B increase Nr4a1 binding on these target genes?

7. The ChIP results are plotted as % enrichment of input. There are very large differences in the % enrichment in the saline control condition across experiments, such as Fig. 1K H3K27me3 enrichment showed less than 0.2% in 1 day and more than 10% in saline group of 28 days. What is the source of this extreme technical variability across different experiments?

8. Inconsistent reporting of behavioral data is used in the figures. In Figure 1, the infusions and DR are reported, and active/inactive spins are reported in the supplement. In contrast, only nose pokes are reported for the cue-seeking experiments. Please report infusions, DR, and inactive for Figure 4.

9. The use of both self-administration model and chronic IP cocaine could possibly explain some of the discrepancies in the findings and should be discussed.

10. Is there a known mechanism for the hypothesized hPTM changes that Nr4a1 produces? Why is it specific to H3K27me3 and not H3K4me3 as in Figure 3I?

11. Does the manipulation in Fig 3 have any effect on H3K27ac? In this figure only 2 histon marks are reported (in contrast to the rest of the figure).

12. The authors conclude that Nr4a1 suppresses "cocaine-induced behavior via downstream target activation in late abstinence". While this may be the case, this conclusion is not warranted based on these current data. Certainly the authors have shown that Nr4a1 is important, but its mechanism has not been demonstrated. In fact, the effect of acute Csn-b in Figure 5F argues against this point. A rephrasing of their conclusions is needed.

13. Similar to #6, in the introduction, "Nr4a1-induced Cartpt and Vmat2 expression is both necessary and sufficient to regulate cocaine behavior" goes too far. Clearly Nr4a1 is important, and it clearly regulates these two genes, however specific regulation has not been demonstrated to be causal for the behaviors.

14. In the results of 4E, was cocaine preference "correlated" with gene expression or simply "associated" with it? Do those values correlate? If so, these data are missing.

15. The method of Csn-B administration is confusing. In the figure legends, the text reads "2x daily, 4 days". However, in the main text, it is described as "repeated (4 injections/day)". Please clarify the experimental procedures.

Reviewer #1 (Remarks to the Author):

This article is attempting to identify potential homeostatic mechanisms in the NAc that are working to reverse drug-induced pathology after cocaine exposure, thereby reducing risk of relapse. They interestingly found that cocaine self-administration transiently activates Nr4a1 in the NAc, subsequently contributing to activation of homeostatic genes during late abstinence. Previously Nr4a1 has been identified as an upstream regulator of gene expression during late abstinence, so this is not the first work to suggest an important role of Nr4a1 in regulating gene expression following cocaine exposure. This paper built on these previous findings with genome wide RNA sequencing and by identifying epigenetic mechanisms for the activation of Nr4a1, namely changes in histone modifications. Furthermore, although Nr4a1 was only enriched after 1 day of abstinence, it was still enriched in the promoters of homeostatic genes such as cartpt and vmat2 at 28 days. The authors then drove Nr4a1 activation to physiologically-relevant levels (matching cocaine IVSA) and find that activation and repression of nr4a1 increased and decreased expression of cartpt and vmat2 respectively. The expression of cartpt and vmat2 when nr4a1 is active (ie with viral transfection or after cocaine IVSA) might be dependent on the enrichment of repressive hPTMs. This nr4a1 induced expression of cartpt and vmat2 could be functionally relevant to decreased cocaine-related behaviors as nr4a1 over-expression, both virally and pharmacologically, decreased cocaine conditioned place preference and viral over-expression reduced cocaine seeking. These results are very compelling and suggest that these homeostatic genes, cartpt and vmat2, could be potential therapeutic targets for addiction, particularly considering there is a potential pharmacological treatment. However, there are some issues with the language and statistics used and a few additional experiments seem necessary to complete this interesting story.

Major:

Statistical issues:

There are several statistical issues in this manuscript. The major issue is the use of t-tests throughout instead of ANOVAs since day can be used as a factor. T-tests after 1 or 28 days of abstinence are used for each gene of interest or hPTM when it would be more appropriate to use ANOVAs. In some instances, the failure to use ANOVAs seems to be favoring t-tests since ANOVAs would result in main effects of cocaine and no effect of # of days. The fact that each group is analyzed by t-test is just not appropriate.

We agree that our manuscript is strengthened by using more rigorous statistical tests. We now use the # of days and treatment as factors in two-way ANOVAs to robustly determine the effects of abstinence and cocaine on gene expression and hPTMs. The statistical methods have been updated to reflect this change (Page 40, Lines 951-969), as well as the figure legends. In all cases, post hoc analysis was used to specify the significant differences between groups, to clarify that ANOVA effects of treatment were within the number of days, or to confirm that there were main effects of treatment.

Also, it is hard to tell whether the t-tests used are comparing the drug or virus treatment to the control group or to 1. The control groups (for example Fig3H) are not shown, but the fold change indicates the control groups are at 1. However, it is unclear whether the analysis is being done against 1 versus treatment or the controls, which although set to 1 would maintain variability. This should be made more clear since control data are not shown. Also, figure captions incorrectly say control is shown in grey.

We thank the reviewer for pointing out this important clarification. In all cases data was compared to controls (e.g. saline, sgRNA-NT, or DMSO). We have stated these comparisons more clearly in the text and figure legends. In several cases we have now added control data, and all control data can be found in the source data file.

A statistic section also needs to be added to the methods of this paper. A lot of these issues would be made clear if the analyses were laid out in one place. That p-values used should be made more explicit. Especially cut-offs used for RNAseq and what constitutes a trend. For RNAseq the p-value appears to be 0.05, which doesn't seem to be stringent enough. A table should be included with the # of differentially expressed genes across stringencies. Furthermore, only one trending effect was presented in the figure and I found the p-value in the figure as .159. What a "trending" value is should be laid out and if the authors include trends that should be clear in the results, what is significant and what is a trend. Furthermore, any other values that fall within that trending p-value should be called a trend as well.

We thank the reviewers for identifying these discrepancies in our manuscript. We have corrected language throughout the manuscript to define significant and not significant data. We have now included a revised statistical methods section (Page 40, Lines 951-969), which states tests conducted for each experiment.

We used several pipelines to identify differentially expressed genes with the goal of increasing reproducibility. Our assumption is that a reproducible change should be identified using a number of different normalizations, and analysis methods. We used a q-value (adjusted p-value) cut off of 0.25 ¹, which is the probability a significant result is a false positive among significant features (genes) rather than the probability of false positives among all null features (p-value) ². Genes satisfying the testing threshold of a q-value < 0.25 across each individual combination for a minimum of 5 (max. 45) combinations were identified as differentially expressed (Page 37, Lines 871-881). Because the number of successes in a sequence of *n* independent experiments follows a binomial distribution, we conclude that our q-value cut off is 0.01 using the equation below, when considering the probability of being identified on 5 out of 45 tests (n=5, k=45, p=0.25).

$$P(x=5) = \binom{n}{k} * p^k * (1-p)^{n-k}$$

To validate our findings, we chose genes across varying levels of significance (q-values and # of algorithms) and 13 out of 13 genes tested were validated in a separate cohort of mice (Supp. Fig 4A-D). Because we validated our findings with subsequent biological verification, a q-value threshold can be phrased as the proportion of significant genes that turn out to be false leads ². Therefore, we would expect to find 1 false positive out of our 44 DEGs at 1-day of abstinence. We have included a table (an individual tab in Table 1) with the number of DEGs across stringencies. Shown below we can see that our method performs similar to a p<0.0001 in NAc.

	Stringency	1-day	28-day
NAc	q < 0.01	44	341
	p < 0.001	55	47
	p < 0.01	179	304
	p < 0.05	894	1117

The authors should mention what post-hoc tests are being performed after significant interactions and whether multiple comparisons are being corrected for.

In all cases, post-hoc test were used to correct for multiple comparisons when appropriate. Details can now be found in the figure legends and revised statistical methods section (Page 40, Lines 951-969).

Conditioned place preference issues:

There are some more major issues with the way the CPP findings are described. I don't disagree that nr4a1 over-expression decreased cocaine preference and repression increases preference, but some things are mentioned that aren't necessarily true. The authors say that the groups spent equal time in both chambers during the pre-test, but that doesn't appear to be true since 40% was spent in the future cocaine-paired side and that indicates a bias. They don't have different biases, which may be the authors point, but this should be made more clear.

We agree with the reviewer that groups do not spend equal time between the drug paired and saline paired chamber during the pre-test session, rather, mice show no significant differences in their preference (cocaine-paired minus saline-paired) for their future drug paired chamber (two-way RM ANOVA, followed Bonferroni multiple comparisons test to compare effects of treatment)^{3,4}. We have changed the text to describe this difference more clearly (Page 10, Lines 214-215).

The authors also say that both groups spent more time in the cocaine-paired chamber, but that doesn't appear to be true and statistics do not show this. If you draw a line at 50% which is no preference for cocaine that is where nr4a1 over-expression is, indicating mice do not spend more time in cocaine than saline-paired chambers. The authors can either change the text to refer to more time post-test compared to pre-test in the cocaine-paired chamber or you can show the data as post-test – pre-test time in cocaine and just indicate textually that the mice still have a preference for the cocaine paired side compared to the pre-test day.

We have now reported the CPP data as a preference score calculated as time spent in the cocaine-paired chamber minus the time spent in the saline-paired chamber, in seconds^{3,4}. Mice were defined as having acquired cocaine CPP when the average cocaine preference score was significantly higher than the preference score at baseline within groups (two-way RM ANOVA, followed Bonferroni multiple comparisons test to compare main effects of session)^{3,4}. We have added more details describing our paradigm in the methods section (Page 35, Lines 831-834) and indicated textually that mice spend more time post-test compared to pretest (Page 10, Lines 215-217). Results show that at a 10 mg/kg dose, following CRISPR-mediated activation of Nr4a1 (dCas9-VP64), mice transfected with either sgRNA-NT or sgRNA -231 spent more time in the cocaine-paired chamber on the post-test when compared to baseline pre-test preference scores (Fig 4B). However, mice transfected with sgRNA -231 spent significantly less time in the drug paired chamber on the post-test than control mice transfected with sgRNA-NT. Bonferroni's multiple comparisons test found a significant effect of session in both NT (10 and 20 mg/kg, $P < 0.0001$) and -231 (10 mg/kg $P = 0.0135$; 20 mg/kg $P = .0034$), indicating that both doses mice transfected with sgRNA -231 developed a preference for the cocaine-paired chamber, when compared to the average baseline preference scores.

It is also critically important to show CPP results at multiple doses. A 10 mg/kg dose for is used for the activation experiment and 5 mg/kg for the repression. The authors call the latter dose a sub-threshold dose, but in our, and other labs experience, this is not the case. Mice still condition to much lower doses of cocaine, just to lesser amounts. If analysis is done by seconds instead of percentages that might help depict the data better. The authors should seriously consider running CPP with at least one other dose. It is important to show that the mice aren't impaired in their

context associated learning (in fig4b) so having a dose where the mice do condition well is important. Also, it would be nice to see a dose where the controls show preference and whether preference is further increased after nr4a1 repression or is equal to controls.

We agree that it is essential to show CPP at multiple doses to determine if Nr4a1 activation impairs context associated learning or if preference is further increased at a dose where controls show preference. We now present the data as a preference score in seconds rather than percentages to show mice spend more time in the cocaine-paired chamber on the post-test when compared to the time spent on the pretest (Page 10, Lines 215-217; Fig 4). At a 20 mg/kg dose, following CRISPR-mediated activation of Nr4a1 (dCas9-VP64), mice transfected with either sgRNA-NT or sgRNA -231 spent more time in the cocaine-paired chamber on the post-test when compared to baseline pre-test preference scores. However, mice transfected with sgRNA -231 spent significantly less time in the drug paired chamber on the post-test (Fig 4B). This demonstrates that Nr4a1 activation reduces cocaine condition place preference at two doses of cocaine. Although mice did not spend significantly more time on the cocaine paired chamber, our data with Csn-B administration prior to cocaine conditioning (Fig 5H) provides additional evidence Nr4a1 activation does not impair context associated learning.

Finally, we performed cocaine CPP at 10 mg/kg to determine if CRISPR-mediated repression of *Nr4a1* (dCas9-KRAB) increased the preference for cocaine at a dose at which control mice show preference for cocaine. At a dose where control mice spend more time in the cocaine paired chamber, results show that Nr4a1 repression prior to cocaine conditioning significantly enhanced preference for cocaine when compared to control (Fig 4D). Preference in the sgRNA-231 group was not further increased beyond that observed at the 5mg/kg dose. However, there was a significant difference in the time spent in the cocaine-paired chamber when comparing the control group at the two doses (data not shown). We have also described the 5 mg/kg dose as a low dose, rather than a sub-threshold dose (Page 10, Line 224).

More issues:

Another major issue seems to generally be why vmat2 is such a large part of this paper. There are many effects that are inconsistent between cartpt and vmat2, some of which suggest that vmat2 could not be contributing to mitigating the behavioral effects seen with activating/repressing nr4a1 etc. One example of this is figure 5b where Csn-B treatment acutely or repeated for 1 day actually decreases vmat2 expression. Also, the authors report a trend for cocaine IVSA to increase vmat2 in RNAseq, but that p-value is a .159. That is much higher than the typical trending p-value of up to 0.1. It seems as though the authors might have found this in their qPCR before RNAseq and were trying to include it in the paper even though the results with vmat2 aren't compelling.

We have removed Vmat2 from the manuscript given the inconsistencies stated below in the reviewer's comments.

The model the authors propose is also a bit confusing, mostly because of the inconsistent results between cartpt and vmat2, but vmat2 is also included on the figure. The authors have included the presence of 27me3 and 4me3 at day 1, which is fine, and then 4me3 and k27ac at 28 days, but the results actually show that is enriched at 1 day for cartpt as well (it also looks close for vmat2). The bigger issue is with me3, since it is only elevated for cartpt (but again at both times not just 28 as in the figure) but not vmat2, so these two genes appear to be regulated differently. Perhaps the authors can emphasize more the change in repression, where there is not only more repression at day1, but less repression at day 28 with some varying things happening to antinational modifications.

We agree with the reviewers that our results provide evidence there are similarities in cocaine-induced histone modifications at *Cartpt* and *Vmat2*, but there may be differences in the way these two genes are regulated. To address this, we have removed *Vmat2* from the manuscript entirely. This allowed us to focus our schematic to on changes in repression at *Cartpt*. We have also revised Figure 6 to show 1.) enrichment of H3K27me3, H3K4me3 and H3K27ac at early abstinence 2.) resolution of H3K27me3 and sustained enrichment of H3K4me3 and H3K27me3 at 28-days of abstinence. We plan to further investigate *Vmat2* and report the results at a later date.

Minor:

Throughout, there are several typos and incorrect grammar. The document should be proofread for these errors.

We have corrected typos and incorrect grammar.

The authors mistakenly call the cocaine and pellet reinforcers rewards, this language should be changed.

We have corrected this language and refer to cocaine and pellets as reinforcers (Lines 88, 232, 804-807).

The authors also say that CPP and ivsa are isomorphic, but they are not. They measure fundamentally different behaviors and many studies have found opposing effects of some manipulation on CPP and IVSA (see for example Nestler lab work with CREB). In fact, the vary paper the authors cited says “CPP measures a learning process that is fundamentally distinct form drug self-administration”. Since both behaviors are present in this paper this is only a textual issue, which should be corrected.

We have corrected this language and describe them has fundamentally different behaviors that measure distinct processes involved in the reinforcing properties of cocaine (Page 9, Lines 209-211).

During cocaine self-administration the number of infusions required for criteria was only 10, but the mice seem to be taking much more and the saline mice seem to be taking at least 10. Why is 10 the criteria?

To remain consistent with previously published literature on the acquisition of cocaine self-administration, we defined acquisition (Page 34, Lines 815-818) as the first of three consecutive sessions during which a mice consumed an average of 5 mg/kg cocaine, or about 10 infusions (8-12 infusions x .7 mg/kg per infusion = 5.6-8.4 mg/kg)⁵⁻⁷. In our lab, mice that meet this criterion continue to reliably self-administer cocaine; and these criteria are similar to intake-based acquisition criterion used by other labs⁵⁻⁷. Mice used for biochemical experiments (i.e. RNA-SEQ and qPCR) were selected to represent the full range of behavior measured. Mice that did not meet these criteria were excluded from behavioral and biochemical analysis.

Also, in the self-administration studies with the viruses # of infusions and discrimination index are not shown. Since the authors specifically discuss reasons why over-expressing nr4a1 prior to self-administration do not affect behavior, seeing those other measures is important.

We have included these other measures to show self-administration does not affect infusions, discrimination, and inactive nose pokes during acquisition or 1-day seeking test (Page 31, Lines 742-764; Supp Fig. 8).

A figure should be included in the supplement showing that to be 1 day and to be 28-day abstinent groups do not differ in their cocaine self-administration.

A supplemental figure has been added that shows 1D and 28D groups used for RNA-Seq do not differ in their cocaine SA (Page 27, Lines 637-661; Supp. Figure 1 C-F). In addition, we have included comparisons of 1D and 28D abstinence groups, in a separate group of mice, which were used for qPCR validation of the RNA-Seq. There are no major differences between these groups (Page 28, Lines 673-690; Supp. Figure 3 A-D).

In figure 4, reinstatement should include the inactive nose poking as well. There could be a different interpretation if inactive nose pokes increase when active nose pokes decrease in the nr4a1 over-expressed group.

We have included all measures including inactivate nose pokes (Page 31, Lines 746-768; Supp. Fig. 9G)

Throughout the figures many axes do not match between 1 and 28 days. I understand that is to see better, but it can make interpreting the findings to be very difficult. Perhaps the authors can use the same axis and use a hash to symbolize skipping space on the y axes so we can see the data compared or use the same larger axis and then include a zoomed in version as an inset figure.

We have changed all graphs to have the same axis across days to allow for the direct comparison across days.

Figure 3 should include a zoomed out image with the virus as well, not just DAPI to show the expression with the morphology intact to make placement clear. Also the closer image provided shows it is from the entire NAc, but this image isn't big enough for that. The position in the section should be more clear, perhaps by including where the commissure is located. Also, colors should be written on the image so it is clear, which color is which protein. There should also be a scale bar included. None of this information is located in the caption either. This image also doesn't show the NeuN staining very well, a better image should be used where red staining (NeuN?) without colocalization can be readily seen.

We have now included an image that shows targeting of NAc core, indicated by the anterior commissure (Page 22, Lines 513-515; Fig. 3E). This image shows colocalization of DAPI (blue), dCas9 (red), NeuN (green), Anterior Commissure (Ac). We have moved the zoomed in image to supplemental figure 5 (Page 30, 715-719).

I believe the key in 5b is wrong.

Figure 5B has been corrected.

The authors should consider running qPCR after the acute Csn-B treatment in figure 5 as well since the behavioral effect was actually larger in that experiment. Also, the authors do not discuss why Vmat2 is not changed after treatment but is increased after treatment during CPP.

We have included qPCR following acute Csn-B treatment and cocaine CPP (Page 26, Lines 632-633; Fig. 5L). We have also discussed why *Cartpt* is not changed after treatment below and in the discussion (Page 15, Lines 352-360).

The work with the Csn-B compound is very interesting since it activates *nr4a1* but also causes these repressive modifications. However, it also still increases expression of *cartpt* and *vmat2*. The authors should discuss how repressive hPTMs would keep *nr4a1* from increasing expression for *cartpt* and *vmat2* until day 28 if they are expressed with this compound and so is the repressive modification.

To better understand how repressive modifications may influence the activation of *Cartpt*, we added new results quantifying H3K4me3, H3K27me3 and H3K27ac after acute administration of Csn-B. At a timepoint when *Cartpt* is activated (6hrs), we found increased H3K4me3 and H3K27me3 enrichment (Fig 5D, F). No significant difference was found in the enrichment of H3K27ac (Fig 5E). In contrast, repeated Csn-B administration causes repression of *Cartpt* expression with no changes in quantified hPTMs. Recent evidence shows that repeated methylphenidate administration increases *Cartpt* expression only after 1-week of abstinence underscoring the role of delayed *Cartpt* regulation⁸. Taking into account increased enrichment of H3K27me3 and H3K4me3 at 1-day of abstinence, we propose the initial presence of repressive modifications may mark genes for subsequent gene activation. Similar mechanisms have been shown in the context neuronal differentiation, such that, genes enriched for bivalent chromatin (activating H3K4me3 and repressive H3K27me3) recruit nucleosome disassembly machinery to facilitate transcription factor binding and subsequent gene activation⁹. Furthermore, genomic regions occupied by significant levels of H3K27me3 are coenriched for histone modifying enzymes including H3K27me3 lysine demethylase (KMA6A), p300 (H3K27ac) and H3K4 methyltransferase mixed lineage leukemia 4 (MLL4)¹⁰⁻¹². It is also important to note that H3K27me3 is not depleted from the *Cartpt* promoter, as was the case following CRISPR-mediated *Nr4a1* activation and during abstinence. Although our data using CRISPR-mediated *Nr4a1* activation is sufficient for *Cartpt* expression, we conclude that the time course of *Nr4a1* expression and psychostimulant induced histone modifications are important factor in delayed activation of *Cartpt* observed during abstinence, evidenced activation of *Cartpt* following cocaine CPP (Fig 5I and L).

Reviewer #2 (Remarks to the Author):

This is a very interesting and well written animal model study on the role of *Nr4a1* transcription factor in cocaine-related reward behavior in the ventral striatum, with key experiments involving in vivo epigenomic editing and *Nr4a1* target gene chromatin profiling.

I have very little to comment, other than Figure 2A, supposedly showing NRBE motif enrichment (for *Nr4a1* binding) at their candidate gene promoters. It is not clear how the authors arrived at that conclusion, I assume they used sonication to prepare their chromatin so the direct binding of *Nr4a1* to a specific motif could not be derived from such type of approach. This issue should be better explained in a revision, or if inconclusive, Figure 2A removed from the paper.

We have removed this figure from the main figures and clarified language describing the location of primers used for qChIP. We agree that this data suggests that *Nr4a1* binds the promoter regions of these genes and our data does not support binding to a particular motif. We have removed any reference to the NRBE binding sites in relation to our *Nr4a1* qChIP data (Page 6, Lines 133).

Reviewer #3 (Remarks to the Author):

The article by Carpenter et al. examines the role of Nr4a1 in regulating gene expression, histone modifications, and cocaine CPP and self-administration behaviors. The authors first examined both 1d vs 28d of abstinence following either self-administration or experimenter-administered injections of cocaine to model incubation of cue-induced drug craving and found differences in both Nr4a1 gene expression and histone modifications. They examined possible downstream regulation of several Nr4a1 target genes, including *Cartpt* and *Vmat2*, at 1 vs 28d abstinence timepoints. They found that upregulation of *Cartpt* and *Vmat2* gene expression associated with depletion of repressive epigenetic marks, H3K27me3. Using a modified Crispr/Cas system or an agonist to regulate Nr4a1 expression and function, the authors show an impact on contingent and non-contingent cocaine behavior. Overall, this is an interesting study with novel results that demonstrate a role for Nr4a1 in cocaine behavior. However, there are numerous weaknesses that need to be addressed.

Detailed comments:

1. Several results are not well supported due to low N's and inconsistent reporting of statistics. Fig. 1J is unconvincing as a negative result given the variability and small Ns. Here the p-value=0.1193 and is reported as negative, but elsewhere in Figure 2B, a p-value = 0.153 is reported with a "#" while in Figure 3H, a p=0.078 is not reported as a statistical trend. Perhaps these are typos (for instance, another example in the legend of 5D a "0.587" is surely supposed to be a "0.0587" as suggested by the "#" in the figure itself). Nonetheless, clarification and increased power in Figure 2 would help to avoid possible false negatives. Similarly, in Fig. 1K, there is a big increase in the mean of enrichment H3K27me3 on 1 day, although there is no significant difference (p=0.0997, but with only n=3). Authors conclude that "We found no difference in repressive modification, H3K27me3". These experiments are underpowered and therefore the conclusions are unconvincing.

We performed a power analysis using the program G*power and used our original data to compute the effect size of our experiments, allowing us to empirically determine sample sizes needed to achieve significant power. We first determined the effect size of our preliminary qChIP data was .7 (mean difference between groups, divided by the standard deviation of control group). We then performed a power analysis with a desired effect size of .7, an alpha error of probability .05, and power of .95. Using these parameters, we calculated a minimum total sample size of 29 or about 7-8 per group. Therefore, we increased our n to 8 for each group. Specifically, for all histone modification ChIP experiments, we added 2 replicates per treatment to the original 28D group. DNA material from our 1D group was depleted, therefore we repeated the 1D ChIP with a total of 8 replicates per treatment group so that each comparison could be run simultaneously on the qPCR machine (as stated in the response to comment 7). These data can be found in Figure 1J-M (Page 19, Lines 438-457) and 2D-G (Page 20, Lines 477-498). Details regarding statistical conclusions can be found in the revised statistical methods section (Page 40, Lines 951-969).

2. The statistical analysis methods are not well described. Please describe the statistical analysis methods in detail. Authors used unpaired two-tailed t test repeatedly in the comparison of more than 3 groups such as Fig 2A, and Fig 3B. It will increase the risk to detect false positive. Please use the appropriate correction for the statistical analyses.

We have addressed statistical concerns above and have added additional replicate experiments for Fig 3B. For experiments listed, we have analyzed the data using a One-way ANOVA for the

comparisons of 3 or more groups. In all cases, we found a statistically significant difference between groups using post-hoc multiple comparisons test. Details can be found in the revised statistical methods section (Page 41, Lines 951-969).

3. Authors concluded that “These findings suggest that abstinence-induced expression of *Cartpt* and *Vmat2* is regulated by sustained *Nr4a1* enrichment and depletion of repressive H3K27me3”. The depletion of H3K27me3 is a possible mechanism of reduction of *Cartpt* and *Vmat2* expression; however, the causal role of sustained *Nr4a1* was not examined. The hPTM changes produced by *Nr4a1* are only correlative, and do not consistently explain the changes observed (e.g. locus-specific *Nr4a1* manipulations vs. agonist on *Cartpt* and *Vmat2* expression).

We agree that our data suggest an association between *Nr4a1* expression/binding and the enrichment of histone modifications. We have toned down language indicating a causal role of *Nr4a1* in the observed mechanism (Page 7, Lines 155-158).

4. Not all of the data support the hPTM conclusions in Figure 6, for example, H3K27ac is increased at 1 day in Fig 2G for *Cartpt*, but not *Vmat2* (though this is another odd statistic as there is $p=.132$, but Ns of only 3 per group). Considering that the qChIP data in Fig 2E shows a significant effect of *Nr4a1* on *Cartpt*, but not *Vmat2*, Figure 6 is too simplistic (see also Fig 2H for *Vmat2* differences). Perhaps these inconsistencies suggest that *Cartpt*, but not *Vmat2*, is important for the behavioral results as both the Cas9 and Can-B changes are consistent for this locus, but are dissociated from the *Vmat2*?

As stated above, we agree with the reviewers that our results provide evidence of similarities in cocaine-induced histone modifications at *Cartpt* and *Vmat2*, but there may be differences in the way these two genes are regulated. To address this, we have removed *Vmat2* from the manuscript entirely, although the results were replicated in separate cohort of mice. This allowed us to simplify our schematic to emphasize changes in repression at *Cartpt*. We have also revised Figure 6 to show 1.) enrichment of H3K27me3, H3K4me3 and H3K27ac at 1-day 2.) resolution of H3K27me3 and sustained enrichment of H3K4me3 and H3K27me3 at 28-days of abstinence.

5. Despite the attempt to use timepoints associated with incubation of drug craving (1 vs 28d abstinence), there is no obvious incubation in the 28d group of mice in Figure 4K. Figure 4H has nose pokes at 1d seeking of about 60, but in a separate group, the nose pokes at 28d seeking are only about 40 for controls. This draws into question the use of these mice in previous figures to study the effects of incubated craving. Why do they not incubate?

We believe that the timeline of stereotaxic surgery has a large effect on both the acquisition of cocaine self-administration and the presentation of incubation. In Figure 4G stereotaxic surgery prior to acquisition blunts responding, when compared to Figure 4J. This may be the case for Figure 4K where stereotaxic surgery is performed prior to the seeking test. In a similar paradigm to ours, enhancement of cocaine seeking was seen in *Vglut2* KO mice when compared to control mice that did not incubate¹³.

Although some studies have observed incubation in mice^{14,15}, several studies suggest session duration¹⁶, dose¹⁶, food training¹⁴ as possible reasons why mice fail to develop incubation under conditions used in rats. Additionally studies have found transient behavioral enhancement at 7-days that declines by 28-days¹⁵. Another possibility is that the incentive properties of contextual memories in mice are less salient when compared to rats¹⁷. Mice used for RNA-Seq received 21 days of cocaine whereas we were limited by the expression pattern of CRISPR-mediated *Nr4a1* expression in the analysis of the effect of *Nr4a1* on self-administration behavior.

None the less several studies have studied these behavioral time points in mice^{13–16,18}. Finally, we are planning to develop a model of incubation in mice to further validate our findings.

6. An acute dose of Csn-b attenuates CPP (Figure 5I), but it does not increase both *Cartpt* and *Vmat2* levels (Figure 5B). Although *Nr4a1* clearly has a role, the mechanism suggested to explain its role is unconvincing at present. Does Csn-B increase *Nr4a1* binding on these target genes?

We have greatly expanded our work and performed qChip for H3K27me3, H3K27ac and H3K4me3 following acute Csn-B administration. We have added discussion of this topic (Page 15, Lines 359-370). At a timepoint when *Cartpt* is activated (6hrs), we found increased H3K4me3 and H3K27me3 enrichment (Fig 5D, F). No significant difference was found in the enrichment of H3K27ac at *Cartpt* (Fig 5E). In contrast, repeated Csn-B administration causes repression of *Cartpt* expression with no changes in quantified hPTMs. Recent evidence shows that repeated methylphenidate administration activates *Cartpt* only after 1-week of abstinence underscoring the role of delayed *Cartpt* regulation⁸. Taking into account increased enrichment of H3K27me3 and H3K4me3 at 1-day of abstinence, we propose the initial presence of repressive modifications may mark genes for subsequent gene activation. Similar mechanisms have been shown in the context neuronal differentiation, such that, genes enriched for bivalent chromatin (activating H3K4me3 and repressive H3K27me3) recruit nucleosome disassembly machinery to facilitate transcription factor binding and subsequent gene activation⁹. Furthermore, genomic regions occupied by significant levels of H3K27me3 are coenriched for histone modifying enzymes including H3K27me3 lysine demethylase (KMA6A), p300 (H3K27ac) and H3K4 methyltransferase mixed lineage leukemia 4 (MLL4)^{10–12}. It is also important to note that H3K27me3 is not depleted from the *Cartpt* promoter, as was the case following CRISPR-mediated *Nr4a1* activation and during abstinence. Although our data using CRISPR-mediated *Nr4a1* activation is sufficient for *Cartpt* expression, we conclude that the time course of *Nr4a1* expression and psychostimulant induced histone modifications are important factor in delayed activation of *Cartpt* observed during abstinence, evidenced by activation of *Cartpt* following cocaine CPP (Fig 5I and L). Although we did not measure *Nr4a1* binding at either time point, future studies could be conducted to further elucidate the mechanism of Csn-B. We have also discussed why *Cartpt* is not changed after treatment below and in the discussion (Page 15, Lines 352-360)

7. The ChIP results are plotted as % enrichment of input. There are very large differences in the % enrichment in the saline control condition across experiments, such as Fig. 1K H3K27me3 enrichment showed less than 0.2% in 1 day and more than 10% in saline group of 28 days. What is the source of this extreme technical variability across different experiments?

Original ChIP experiments were conducted at different timepoints which led to differences in the amount of tissue, antibody lot, total chromatin in the input and equipment used (qPCR machine). We believe that these differences contributed to the large differences in percent input across experiments. Additionally, we identified a mistake in our calculation of percent input given we took different amounts of input (5% vs 10%) This mistake does not change results, per se, given that % input is a relative measure and calculated the same across treatment and control groups. However, we rigorously replicated ChIP experiments using exactly the same amount of tissue, same antibody lots, and total chromatin for input to decrease variability. We generated replicate *Nr4a1* ChIP data (n=6) at both time points and 1-day histone ChIP data (n=8). We used the parameters above to generate original 28D histone ChIP data (n=6) and thus combined original 28D histone data with replicate 28D histone ChIP data (replicate n=2, total n = 8;). We validated our findings and reduced these large differences across experiments by careful consideration of technical variables. These data can be found in Figure 1J-M (Page 19, Lines 433-452) and 2D-G (Page 20, Lines 438-457).

8. Inconsistent reporting of behavioral data is used in the figures. In Figure 1, the infusions and DR are reported, and active/inactive spins are reported in the supplement. In contrast, only nose pokes are reported for the cue-seeking experiments. Please report infusions, DR, and inactive for Figure 4.

These measures have now been included in the supplemental figures 8 and 9.

9. The use of both self-administration model and chronic IP cocaine could possibly explain some of the discrepancies in the findings and should be discussed.

We acknowledge that there are differences between i.p and iv cocaine that could contribute to discrepancies (Page 14, Lines 340-349)^{19,20}. The fact that we test the effect of *Nr4a1* activation on both paradigms provides evidence that the mechanism of *Nr4a1* activation is conserved between the two models. We identified consistent *Nr4a1*/*Cartpt* activation in response to investigator administered and self-administered cocaine. We also observe the same behavioral phenotype following CRISPR-mediated *Nr4a1* activation on cocaine CPP and self-administration. We have added details to the discussion regarding these models (Page 14, Lines 340-349).

10. Is there a known mechanism for the hypothesized hPTM changes that *Nr4a1* produces? Why is it specific to H3K27me3 and not H3K4me3 as in Figure 3I?

We hypothesize that cocaine induced H3K27me3 enrichment acts in an epistatic manner to repress *Nr4a1* target gene expression in the presence of H3K4me3 and H3K27ac²¹. Recent *Nr4a1* and H3K27ac ChIP-seq data suggest that *Nr4a1* creates large scale open chromatin regions, such that, *Nr4a1* binding positively correlates with H3K27ac at activated genes²². Depletion of the H3K27me3 is associated with the enrichment of acetyltransferase, p300, a protein known to interact with *Nr4a1* to coordinate the enrichment of H3K27ac^{23,24}. Furthermore, genomic regions occupied by significant levels of H3K27me3 are coenriched for histone modifying enzymes including H3K27me3 lysine demethylase (KMA6A), p300 (H3K27ac) and H3K4 methyltransferase mixed lineage leukemia 4 (MLL4)¹⁰⁻¹². We show that *Nr4a1* activation depletes H3K27me3 and enriches H3K27ac facilitating activation of *Cartpt* during abstinence (Page 22, Lines 534-538; Fig. 3 J,K)²⁵. We have added discussion of this mechanism (Page 13, Lines 311-320).

11. Does the manipulation in Fig 3 have any effect on H3K27ac? In this figure only 2 histone marks are reported (in contrast to the rest of the figure).

Yes, CRISPR-mediated *Nr4a1* activation enriched H3K27ac at the *Cartpt* promoter. We have now included the data in the revised manuscript (Fig. 3K).

12. The authors conclude that *Nr4a1* suppresses “cocaine-induced behavior via downstream target activation in late abstinence”. While this may be the case, this conclusion is not warranted based on these current data. Certainly, the authors have shown that *Nr4a1* is important, but its mechanism has not been demonstrated. In fact, the effect of acute *Csn-b* in Figure 5F argues against this point. A rephrasing of their conclusions is needed.

This conclusion has been rephrased in the discussion (Page 16, Lines 374-379).

13. Similar to #6, in the introduction, “*Nr4a1*-induced *Cartpt* and *Vmat2* expression is both necessary and sufficient to regulate cocaine behavior” goes too far. Clearly *Nr4a1* is important,

and it clearly regulates these two genes, however specific regulation has not been demonstrated to be causal for the behaviors.

This conclusion has been rephrased (Page 4, 85-89).

14. In the results of 4E, was cocaine preference “correlated” with gene expression or simply “associated” with it? Do those values correlate? If so, these data are missing.

Activation/Repression of *Cartpt* expression was associated with cocaine preference and no significant correlations were found (Page 10, Lines 221, 225).

15. The method of Csn-B administration is confusing. In the figure legends, the text reads “2x daily, 4 days”. However, in the main text, it is described as “repeated (4 injections/day)”. Please clarify the experimental procedures.

We have clarified language regarding Csn-B administration (Page 25, line 592).

List of changes

Page 4: Removed data and discussion of *Vmat2* from the manuscript (Reviewer 1 and 3)

Page 4: Rephrased conclusions on the effect of *Nr4a1* on cocaine-mediated behaviors (Reviewer 3; Lines 84-89).

Page 5: Added new results showing 1D and 28D RNA-seq groups are similar in cocaine SA (Reviewer 1; Supp. Fig. 1C-F, Line 100).

Page 5: Added new statistics using 2-way ANOVA (Reviewer 1 & 3; Figure 1G, Line 110)

Page 5: Added new results showing 1D and 28D qPCR biological replicate groups are similar in cocaine SA (Reviewer 1; Supp. Fig. 2A-D, Lines 110).

Page 6: Added new statistics using 2-way ANOVA (Reviewer 1 & 3; Figure 1I, Page 5 Line 123)

Page 6: Added new results of *Nr4a1* ChIP in new biological samples and new statistics using 2-way ANOVA (Reviewer 1 & 3; Fig. 1J, Line 124).

Page 6: Added new results of H3K27me3 in new biological samples and new statistics using 2-way ANOVA (Reviewer 1 & 3; Fig. 1K, Line 126)

Page 6: Added new results of H3K27ac in new biological samples and new statistics using 2-way ANOVA (Reviewer 1 & 3; Fig. 1L, Line 127)

Page 6: Added new results of H3K27ac in new biological samples and new statistics using 2-way ANOVA (Reviewer 1 & 3; Fig. 1M, Line 127)

Page 6: Removed reference to *Nr4a1* binding at NRBE sites (Reviewer 2; Supp. Fig. 5, Line 133)

Page 7: Added new statistics using 2-way ANOVA (Reviewer 1 & 3; Figure 2A, Lines 133-140)

Page 7: Added new statistics using 2-way ANOVA (Reviewer 1 & 3; Figure 2C, Lines 145-148)

Page 7: Added new results of Nr4a1 ChIP in new biological samples and new statistics using 2-way ANOVA (Reviewer 1 & 3; Fig. 2D, Lines 148-149).

Page 7: Added new results of H3K27me3 in new biological samples and new statistics using 2-way ANOVA (Reviewer 1 & 3; Fig. 2E, Lines 150-151)

Page 7: Added new results of H3K27ac in new biological samples and new statistics using 2-way ANOVA (Reviewer 1 & 3; Fig. 2F, Lines 151-152)

Page 7: Added new results of H3K4me3 in new biological samples and new statistics using 2-way ANOVA (Reviewer 1 & 3; Fig. 2G, Lines 151-152)

Page 7: Rephrased conclusions related to Nr4a1 and associated histone modification at *Cartpt* during abstinence (Reviewer 1 & 3; Lines 155-158)

Page 8: Added additional replicate experiments and new statistics using 1-way ANOVA (Reviewer 3; Fig. 3B, Lines 167-170)

Page 8: Added new images including the anterior commissure, included scaled and details pertaining to the staining. Included zoomed in images to the supplement (Reviewer 1; Fig. 3E, Supp. 6, Lines 185-189)

Page 9: Added new statistics using 1-way ANOVA (Reviewer 3; Figure 3F, Lines 192-194)

Page 9: Added new statistics using 2-way ANOVA (Reviewer 3; Figure 3I, Lines 197-199)

Page 9: Added new data H3K27ac qChIP data following CRISPR-mediated activation (Reviewer 3; Figure 3I, Lines 199-200)

Page 9: Rephrased statements regarding CPP and SA (Reviewer 1; Lines 210-212, Lines 209-211)

Page 10: Clarified language describing differences in preference during pretest (Reviewer 1; Lines 217-218, Lines 214-215)

Page 10: Changed text to refer to more time spent on the cocaine paired chamber when compared to pretest (Reviewer 1; Lines 215-217)

Page 10: Represented CPP data as preference score in seconds (Reviewer 1 Fig. 4A, D, Lines 215-220)

Page 10: Added new dCas9-VP64 CPP data 20 mg/kg (Reviewer 1; Fig 4C, Lines 219-220)

Page 10: Clarified repression and enhancement of cocaine CPP is associated with regulation of *Cartpt*, not correlated (Reviewer 3; Lines 221, 225)

Page 10: Added new dCas9-KRAB CPP data 10 mg/kg (Reviewer 1; Fig 4G, Lines 224-225)

Page 10: Changed references to reward to reinforcers (Reviewer 1; Lines 231)

Page 10: Added figures of all measures of SA (Reviewer 1 & 3; Supp. 8 and 9, Line 230-237)

Page 11: Added new qChIP data following acute Csn-B administration (Reviewer 1 & 3; Fig. 5D-F, Lines 248-251)

Page 11: Added new qPCR data following acute Csn-B CPP (Reviewer 1; Fig. 5L, Lines 254-255)

Page 12: Updated discussion with new results (Reviewer 1 & 3; Lines 260-276)

Page 13: Revised Figure 6 to included hPTMs at *Cartpt* during early and late abstinence (Reviewer 1 & 3; Fig. 6, Lines 305-308)

Page 13: Added discussion on the known mechanism for the hypothesized hPTMs Nr4a1 induces (Reviewer 3; Lines 313-320)

Page 14: Added references that use 28-days abstinence time points in mice (Reviewer 3; Lines 338-339)

Page 14: Added discussion of difference between voluntary and involuntary drug exposure paradigms (Reviewer 3; Lines 340-344)

Page 15: Added discussion on the difference in Csn-B administration and CRISPR-mediated Nr4a1 activation. Added discussion on Csn-B and repressive histone modifications (Reviewer 1 & 3; Lines 352-360)

Page 16: Rephrased conclusions (Reviewer 3; Lines 373-379)

Page 25: Clarified Csn-B paradigm (Reviewer 3; line 592)

Page 34: Changed references to reward to reinforcers (Reviewer 1; line 805, 823)

Page 35: Updated methods for cocaine CPP (Reviewer 1; Lines 831-844)

Page 37: Updated methods for RNA-Seq analysis (Reviewer 1; Lines 871-881)

Page 40: Added revised statistical methods section to methods (Reviewer 1; Lines 951-969)

Response References

1. Mulligan, G. *et al.* Gene expression profiling and correlation with outcome in clinical trials of the proteasome inhibitor bortezomib. *Blood* **109**, 3177–3188 (2007).
2. Storey, J. D. & Tibshirani, R. Statistical significance for genomewide studies. *Proc. Natl. Acad. Sci.* **100**, 9440–9445 (2003).
3. Burgdorf, C. E. *et al.* Extinction of Contextual Cocaine Memories Requires Ca v 1.2 within D1R-Expressing Cells and Recruits Hippocampal Ca v 1.2-Dependent Signaling Mechanisms. *J. Neurosci.* **37**, 11894–11911 (2017).
4. Tropea, T. F., Kosofsky, B. E. & Rajadhyaksha, A. M. Enhanced CREB and DARPP-32 phosphorylation in the nucleus accumbens and CREB, ERK, and GluR1 phosphorylation in the dorsal hippocampus is associated with cocaine-conditioned place preference

- behavior. *J. Neurochem.* **106**, 1780–1790 (2008).
5. Carroll, M. E. & Lac, S. T. Acquisition of IV amphetamine and cocaine self-administration in rats as a function of dose. *Psychopharmacology (Berl)*. **129**, 206–214 (1997).
 6. Mantsch, J., Ho, A., Schlussman, S. & Kreek, M. Predictable individual differences in the initiation of cocaine self-administration by rats under extended-access conditions are dose-dependent. *Psychopharmacology (Berl)*. **157**, 31–39 (2001).
 7. Mandt, B. H. & Zahniser, N. R. Low and high cocaine locomotor responding male Sprague–Dawley rats differ in rapid cocaine-induced regulation of striatal dopamine transporter function. *Neuropharmacology* **58**, 605–612 (2010).
 8. Jayanthi, S. *et al.* Methamphetamine Induces TET1- and TET3-Dependent DNA Hydroxymethylation of Crh and Avp Genes in the Rat Nucleus Accumbens. *Mol. Neurobiol.* **55**, 5154–5166 (2018).
 9. Gao, Y., Gan, H., Lou, Z. & Zhang, Z. Asf1a resolves bivalent chromatin domains for the induction of lineage-specific genes during mouse embryonic stem cell differentiation. *Proc. Natl. Acad. Sci.* **115**, E6162–E6171 (2018).
 10. Kim, J.-H. *et al.* UTX and MLL4 Coordinately Regulate Transcriptional Programs for Cell Proliferation and Invasiveness in Breast Cancer Cells. *Cancer Res.* **74**, 1705–1717 (2014).
 11. Dhar, S. S. *et al.* MLL4 Is Required to Maintain Broad H3K4me3 Peaks and Super-Enhancers at Tumor Suppressor Genes. *Mol. Cell* **70**, 825–841.e6 (2018).
 12. Wang, S.-P. *et al.* A UTX-MLL4-p300 Transcriptional Regulatory Network Coordinately Shapes Active Enhancer Landscapes for Eliciting Transcription. *Mol. Cell* **67**, 308–321.e6 (2017).
 13. Alsio, J. *et al.* Enhanced Sucrose and Cocaine Self-Administration and Cue-Induced Drug Seeking after Loss of VGLUT2 in Midbrain Dopamine Neurons in Mice. *J. Neurosci.* **31**, 12593–12603 (2011).
 14. Halbout, B., Bernardi, R. E., Hansson, A. C. & Spanagel, R. Incubation of Cocaine Seeking following Brief Cocaine Experience in Mice Is Enhanced by mGluR1 Blockade. *J. Neurosci.* **34**, 1781–1790 (2014).
 15. Nugent, A. L., Anderson, E. M., Larson, E. B. & Self, D. W. Incubation of cue-induced reinstatement of cocaine, but not sucrose, seeking in C57BL/6J mice. *Pharmacol. Biochem. Behav.* **159**, 12–17 (2017).
 16. Terrier, J., Lüscher, C. & Pascoli, V. Cell-Type Specific Insertion of GluA2-Lacking AMPARs with Cocaine Exposure Leading to Sensitization, Cue-Induced Seeking, and Incubation of Craving. *Neuropsychopharmacology* **41**, 1779–1789 (2016).
 17. Tirelli, E., Laviola, G. & Adriani, W. Ontogenesis of behavioral sensitization and conditioned place preference induced by psychostimulants in laboratory rodents. *Neurosci. Biobehav. Rev.* **27**, 163–178 (2003).
 18. Walker, D. M. *et al.* Cocaine Self-administration Alters Transcriptome-wide Responses in the Brain's Reward Circuitry. *Biol. Psychiatry* (2018). doi:10.1016/j.biopsych.2018.04.009
 19. Ma, F., Falk, J. L. & Lau, C. E. Within-subject variability in cocaine pharmacokinetics and pharmacodynamics after intraperitoneal compared with intravenous cocaine administration. *Exp. Clin. Psychopharmacol.* **7**, 3–12 (1999).
 20. Pan, H.-T., Menacherry, S. & Justice, J. B. Differences in the Pharmacokinetics of Cocaine in Naive and Cocaine-Experienced Rats. *J. Neurochem.* **56**, 1299–1306 (1991).
 21. Bernstein, B. E. *et al.* A Bivalent Chromatin Structure Marks Key Developmental Genes in Embryonic Stem Cells. *Cell* **125**, 315–326 (2006).
 22. Liu, X. *et al.* Genome-wide analysis identifies NR4A1 as a key mediator of T cell dysfunction. *Nature* **567**, 525–529 (2019).
 23. Pasini, D. *et al.* Characterization of an antagonistic switch between histone H3 lysine 27 methylation and acetylation in the transcriptional regulation of Polycomb group target

- genes. *Nucleic Acids Res.* **38**, 4958–4969 (2010).
24. Duren, R. P., Boudreaux, S. P. & Conneely, O. M. Genome wide mapping of NR4A binding reveals cooperativity with ETS factors to promote epigenetic activation of distal enhancers in acute myeloid leukemia cells. *PLoS One* **11**, 1–20 (2016).
 25. Sinha, N., Biswas, A., Nave, O., Seger, C. & Sen, A. Gestational Diabetes Epigenetically Reprograms the Cart Promoter in Fetal Ovary, Causing Subfertility in Adult Life. *Endocrinology* **160**, 1684–1700 (2019).

REVIEWERS' COMMENTS:

Reviewer #1 (Remarks to the Author):

The authors have addressed my prior concerns.

Reviewer #2 (Remarks to the Author):

The authors have properly responded to my previous comment, and it appears to me that they also adequately responded to previous comments and suggestions by the other two Reviewers. I have no additional comments.

Reviewer #3 (Remarks to the Author):

The authors have addressed my concerns adequately.

Reviewer #1 (Remarks to the Author):

The authors have addressed my prior concerns.

Reviewer #2 (Remarks to the Author):

The authors have properly responded to my previous comment, and it appears to me that they also adequately responded to previous comments and suggestions by the other two Reviewers. I have no additional comments.

Reviewer #3 (Remarks to the Author):

The authors have addressed my concerns adequately.

We thank the reviewers for their efforts and support of our manuscript